# CHB: A Diagnostic Toolkit for Hardness-Aware Clustering Evaluation

**Walid Durani** [1 2]   **Philipp Jahn** [1 2]   **Collin Leiber** [3]   **David B. Hoffmann** [1]   **Thomas Seidl** [1 2]   **Claudia Plant** [4 5]
**Christian Böhm** [4]

## Abstract

Clustering methods are commonly compared through leaderboards that collapse performance into a single aggregated ranking. Such summaries do not reveal why methods succeed, which data properties align with failure, and how conclusions shift under representation changes and realistic tuning constraints. We present the Clustering Hardness Benchmark (CHB), a diagnostic toolkit for hardness-aware clustering via external evaluation. CHB maps each dataset to an interpretable hardness fingerprint capturing (i) separation, (ii) cohesion and scale heterogeneity, and (iii) topology. Using this diagnostic space, CHB evaluates clustering algorithms under standardized default configurations and budgeted hyperparameter tuning. Conditioning results on hardness coordinates turns comparison into diagnosis: across a broad range of datasets and their representations, CHB reveals reproducible structural regimes, uncovers regime-dependent ranking across method families, and surfaces robustness signatures, including topology-linked breakdowns. CHB further enables representation auditing by attributing gains to measurable shifts in the hardness fingerprint rather than just external performance changes. We release CHB as an open, extensible artifact for evaluating new clustering methods and embeddings within a shared diagnostic framework.

## 1. Introduction

Clustering is central to exploratory analysis and downstream systems such as retrieval and recommendation, and it is in-

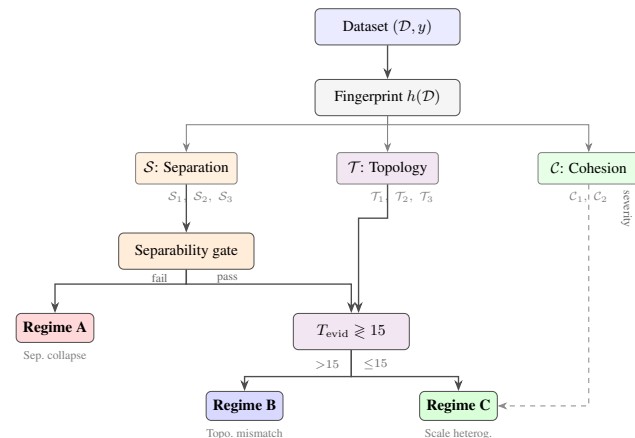

*Figure 1.* CHB workflow. Given a dataset $\mathcal{D}$ with reference labels $y$, CHB computes a fingerprint along three interpretable axes and assigns a hardness regime via a separability gate followed by a blob-calibrated topology test. Within Regime C, cohesion ($\mathcal{C}_1, \mathcal{C}_2$) serves as a severity axis that quantifies residual difficulty from scale and density heterogeneity.

creasingly used to evaluate learned representations. Yet benchmark results are often hard to interpret. The dominant practice is to report external agreement metrics—Adjusted Rand Index (ARI) (Hubert & Arabie, 1985), Normalized Mutual Information (NMI) (Strehl & Ghosh, 2002), or unsupervised clustering accuracy (ACC) (Yang et al., 2010)— and average them across diverse dataset suites. In effect, progress is reported as a single leaderboard rank on a fixed data collection. This summary hides the main source of variation: clustering is constrained by *structure*. Different datasets fall into different hardness regimes and method families encode different assumptions about geometry and connectivity. As a result, rankings can flip across regimes. Representation learning compounds the issue by actively modifying the space in which clustering is performed, shifting difficulty and potentially invalidating (or satisfying) algorithmic assumptions (Ackerman & Ben-David, 2009; Allaoui et al., 2020; Asyaky & Mandala, 2021). Because leaderboards rarely report regimes, diagnose rank reversals, or attribute gains to specific failure modes (e.g., separability collapse vs. scale mismatch), they conflate genuine methodological advances with incidental improvements and favorable benchmark selection. We introduce the Clus-

[1]Data Mining and AI Group, LMU Munich, Munich, Germany [2]Munich Center for Machine Learning, Munich, Germany (MCML) [3]Department of Computer Science, Aalto University, Espoo, Finland [4]Faculty of Computer Science, University of Vienna, Vienna, Austria [5]ds:UniVie, University of Vienna, Vienna, Austria. Correspondence to: Walid Durani <durani@dbs.ifi.lmu.de>.

*Proceedings of the 43$^{rd}$ International Conference on Machine Learning*, Seoul, South Korea. PMLR 306, 2026. Copyright 2026 by the author(s).

tering Hardness Benchmark **CHB**, a post-hoc diagnostic framework that advances clustering evaluation from aggregate score-keeping to mechanism-level analysis. Given reference labels—standard in benchmark suites under external evaluation protocols (Leiber et al., 2023; Zhou et al., 2025)—CHB computes a *hardness fingerprint* which embeds each dataset into an interpretable space spanned by three axes: *separability*, *cohesion heterogeneity*, and *topological structure*. Conditioning external evaluation results on these hardness coordinates transforms leaderboards into explanations. CHB isolates reproducible structural regimes, surfaces regime-dependent rank reversals across algorithm families, and enables *representation auditing*—attributing performance gains to measurable shifts along specific hardness axes rather than opaque metric deltas. Figure 1 summarizes the CHB pipeline.

**What CHB enables (Q1–Q3).** CHB reframes clustering evaluation as a causal diagnosis: not just *which method wins*, but *what structural constraint is binding* on each instance. It does so via three coupled analyses that share the same hardness coordinates (separation, cohesion, topology).

**Q1: Regimes.** From descriptors alone, CHB induces stable hardness regimes. The key split is a separability gate: when it fails, performance differences largely collapse; when it holds, difficulty persists for two distinct reasons—either topology/geometry fights "blob-like" assumptions, or heterogeneous scale makes any single global resolution brittle.

**Q2: Mechanism hierarchy.** Regime conditioning turns scattered outcomes into a mechanism story. Separability determines whether clustering is even learnable; conditional on that, cohesion predicts systematic merge–split instability for single-resolution methods; and topology acts as a high-precision discriminator that isolates family-specific failure modes and anticipates regime-dependent ranking flips.

**Q3: Representation auditing.** Finally, CHB evaluates representations by *how they move the instance in hardness space*. Large gains align with shifts that unlock local evidence (separability rescue) and reduce topological entanglements; changes that only "smooth mismatch" without fixing separability tend to help selectively rather than universally.

## 2. Related Work

**Clustering benchmarks and meta-evaluation.** Benchmark suites and automated platforms (Wiwie et al., 2015; Gagolewski, 2022) improve reproducibility but typically report dataset-level aggregates that provide limited explanations of structural failure modes. We instead analyze dataset–representation instances in an interpretable hardness space for regime-conditioned comparison, complementing meta-evaluation and algorithm-selection lines such as ISA (Fernandes et al., 2021) and meta-learning for clustering ranking (De Souto et al., 2008; Soares et al., 2009) by using fixed, theory-motivated descriptors as diagnostic covariates

rather than correlational features for black-box selection.

**Separability indices and data-complexity measures.** Internal validation and clusterability proxies—including DBCV (Moulavi et al., 2014), DCSI (Gauss et al., 2023), and data-complexity measures (Ho & Basu, 2002; Lorena et al., 2019)—provide scalar difficulty indicators but may conflate distinct failure mechanisms that require different algorithmic responses. CHB preserves interpretability while avoiding this conflation.

**Topology in clustering and topology-aware benchmarks.** Persistent homology summarizes point-cloud topology and underlies topology-aware clustering and benchmarks (Ghrist, 2008; Carriere et al., 2018). Existing work, however, does not explain cross-family clustering behavior in general-purpose benchmarks, where topology interacts with overlap, hubness, and density heterogeneity. We use lightweight persistent-homology summaries as hardness indicators, treating topology as an explanatory covariate for regime-dependent failures rather than a task-specific objective.

**Representation learning and unsupervised AutoML.** Embedding benchmarks such as MTEB (Muennighoff et al., 2023) and deep clustering evaluations report external metrics without attributing gains to changes in separability, mismatch, or scale, while unsupervised AutoML and meta-learning methods (Xu & Wunsch, 2005; De Souto et al., 2008; Soares et al., 2009; Vendramin et al., 2010; Liu et al., 2021; Aljoud et al., 2025) focus on algorithm selection rather than diagnosis—even when using modern representations (e.g., CLIP in DCMatch (Aljoud et al., 2025)). In contrast, CHB enables representation auditing via interpretable structural axes and standardized compute tracks, supporting hardness-aware, explanatory analysis.

**CHB in direct comparison to clustering evaluation.** Prior benchmarks standardize datasets, metrics, and protocols across classical, deep, and representation-based clustering methods (Lu et al., 2024; Ren et al., 2025). However, they treat clustering difficulty and deep clustering method performance as empirical outcomes rather than consequences of measurable data structure, offering little explanation of why datasets are hard, when deep methods help or fail, or how embeddings alter clusterability (Asyaky & Mandala, 2021). CHB closes this gap by introducing interpretable hardness descriptors and organizing results into structural regimes, enabling systematic explanations of algorithm behavior and representation effects. More broadly, von Luxburg et al. (2012) argue that clustering evaluation cannot be divorced from end-use context and caution against treating label agreement as a universal quality measure. CHB operates within this label-based protocol—the dominant setting in current benchmarks—not to endorse it as context-free, but to make it more transparent by surfacing the structural vari-

ation that leaderboard scores obscure.

## 3. The CHB Framework

Clustering performance is regime-dependent: methods succeed when their implicit recoverability assumptions align with the geometry induced by a dataset and its representation. CHB is an *external-evaluation* framework that makes this dependence measurable by embedding each *dataset–representation pair* into an interpretable *hardness space*, and then conditioning standard external outcomes (NMI) on position in that space.

**CHB space/ Hardness fingerprint.** For a dataset $\mathcal{D}$, CHB computes a hardness fingerprint

$$\mathbf{h}(\mathcal{D}) = \big[ h_{\mathcal{SEP}}(\mathcal{D}) \,;\, h_{\mathcal{COH}}(\mathcal{D}) \,;\, h_{\mathcal{TOPO}}(\mathcal{D}) \big] \quad (1)$$

where each block targets a distinct family of failure modes (Table 1): (i) **separation** (is there reliable local evidence for membership?), (ii) **cohesion** (do clusters share a compatible internal scale and regularity?), and (iii) **topology** (does multiscale connectivity—chains, bridges, loops, voids—complicate recovery?). All coordinates are designed to be lightweight, comparable across datasets, and estimable at benchmark scale. Conditioning external performance on $\mathbf{h}(\mathcal{D})$ yields mechanism-level diagnosis and enables representation auditing as movement in CHB space.

### 3.1. Fundamental clustering failure modes

Clustering difficulty is mechanism-dependent: different datasets violate different implicit assumptions, so method rankings can legitimately invert across instances. We therefore adopt a *separability-first hierarchy*. When *separability* fails, broad failure is expected: without identifiable local neighborhoods, neither prototype-based nor graph-based objectives have a reliable bootstrap signal (Balcan et al., 2009; Kumar & Kannan, 2010; Von Luxburg, 2007). This role of separability reflects margin- and stability-style conditions under which clustering becomes tractable (Balcan et al., 2009; Kumar & Kannan, 2010). Separability, however, is only a *gate*—necessary but not sufficient. Even with usable margins, methods can fail through inductive-bias mismatch. For example, $k$-means minimizes within-cluster variance and favors compact, single-center structures (MacQueen, 1967), while other methods impose a *single global resolution* (e.g., DBSCAN's $\varepsilon$) (Ester et al., 1996). Accordingly, we treat hardness as a *multi-mechanism* structure. Conditional on separability, failures recur in two modes: **(i) Scale/cohesion mismatch.** A global resolution becomes brittle under non-exchangeable density or spread (core–halo structure, heavy tails, imbalance) (Ester et al., 1996; Comaniciu & Meer, 2002; Campello et al., 2013), and elongation induces chaining and merge–split instability (classically in single-linkage) (Sibson, 1973). We capture

*Table 1.* Mapping between theoretical clustering conditions, CHB metrics, and resulting failure regimes.

| Theoretical condition | CHB metrics | Failure regime |
|---|---|---|
| Local identifiability | $\mathcal{S}_1, \mathcal{S}_2, \mathcal{S}_3$ | Separation collapse |
| Structural alignment | $\mathcal{T}_1, \mathcal{T}_2, \mathcal{T}_3$ | Topological mismatch |
| Scale compatibility | $\mathcal{C}_1 \mathcal{C}_2$ | Multi-scale heterogeneity |

*Table 2.* CHB fingerprint $h(\mathcal{D}) = (\mathcal{S}; \mathcal{C}; \mathcal{T})$. Per-cluster values are aggregated via cluster-size-weighted 10% trimmed mean; scale grids resolved by median first.

| | Description | Defaults |
|---|---|---|
| SEPARATION — *local identifiability* | | |
| $\mathcal{S}_1\uparrow$   Overlap | Cross-label $k$NN fraction | $k \in \{10, 20, 40\}$ |
| $\mathcal{S}_2\uparrow$   Hubness | Hub-mediated shortcut ratio | $k \in \{10, 20, 40\}$ |
| $\mathcal{S}_3\downarrow$   Margin | Gap / within-cluster scale | $q = 0.25, \; p = 3$ |
| COHESION — *scale compatibility* | | |
| $\mathcal{C}_1\uparrow$   Density | Within + across density spread | $k \in \{10, 15, 20\}$ |
| $\mathcal{C}_2\uparrow$   Elongation | $\lambda_1/\mathrm{mean}(\lambda_{2:})$ | cov. spectrum |
| TOPOLOGY — *multiscale structure* (resampled PH, $\tau = 0.01$) | | |
| $\mathcal{T}_1\uparrow$   PH$_0$ | MST-edge persistence sum | $m_0 = 800$ |
| $\mathcal{T}_2\uparrow$   PH$_1$ | Loop lifetime sum | $m_1 = 400$ |
| $\mathcal{T}_3\uparrow$   PH$_2$ | Void lifetime sum | $m_2 = 384, \; L = 128$ |

$\uparrow$ = higher is harder;   $\downarrow$ = lower is harder (thinner margins).

this axis with $(\mathcal{C}_1, \mathcal{C}_2)$. **(ii) Topology Mismatch.** Multiscale connectivity (chains/bridges) and persistent loops/voids violate compact-blob intuitions (Hartigan, 1975; Chaudhuri & Dasgupta, 2010; Ghrist, 2008; Edelsbrunner & Harer, 2010). We capture this axis with persistent-homology (PH)-based indicators $(\mathcal{T}_1 - \mathcal{T}_3)$: PH0 reflects delayed merging and the operative connectivity scale, while PH1/PH2 capture loop/void structure (Ghrist, 2008; Edelsbrunner & Harer, 2010). These diagnostics are not intended as recoverability criteria, but as performance-free indicators of the dominant failure mode for each dataset instance. We summarize the resulting CHB hardness fingerprint below and give full formulas in Table 2, 3, and in Appendix B.

### 3.2. Core CHB Metrics

#### 3.2.1. SEPARATION: LOCAL IDENTIFIABILITY AND ROBUSTNESS.

Separation captures whether local geometric structure provides reliable evidence for cluster membership. Many clustering guarantees implicitly require some form of *local identifiability*: small neighborhoods should be mostly label-consistent and not dominated by spurious shortcuts. Furthermore, inter-cluster gaps should be sufficiently robust to perturbations. (Von Luxburg, 2007; Awasthi et al., 2012; Bilu & Linial, 2012)

*Table 3.* Formal definitions and default parameterization of $h(\mathcal{D}) = (\mathcal{S}; \mathcal{C}; \mathcal{T})$. Per-cluster values are aggregated via cluster-size-weighted 10% trimmed mean; scale grids resolved by median first.

| Definition (per cluster $c$, then aggregated) | Defaults |
|---|---|
| SEPARATION — *local identifiability* | |
| $\mathcal{S}_1 := \operatorname{med}_{k \in \mathcal{K}} \frac{1}{|C_c|} \sum_{i \in C_c} \frac{1}{k} \sum_{j \in N_k(i)} \mathbf{1}[y_j \neq y_i]$ | $\mathcal{K} = \{10, 20, 40\}$ |
| $\mathcal{S}_2 := \operatorname{med}_{k \in \mathcal{K}} \frac{1}{|C_c|} \sum_{v \in C_c} h_k^{\mathrm{cross}}(v)/h_k(v)$ | $\mathcal{K} = \{10, 20, 40\}$ |
| $\mathcal{S}_3 := \operatorname{med}_{p \in \mathcal{P}} Q_q\big(\{\delta_p^{(c)}(i)/s_c\}_{i \in C_c}\big)$ | $q = 0.25$, $\mathcal{P} = \{3\}$ |
| COHESION — *scale compatibility* | |
| $\mathcal{C}_1 := \operatorname{med}_{k \in \mathcal{K}_D} \big[\mathrm{spread}_{c,k} + \max(\tilde{\mu}_k - \mu_{c,k}, 0)\big]$ | $\mathcal{K}_D = \{10, 15, 20\}$ |
| $\mathcal{C}_2 := \lambda_1^{(c)} / \frac{1}{r_c - 1} \sum_{j=2}^{r_c} \lambda_j^{(c)}$ | cov. spectrum |
| TOPOLOGY — *multiscale structure* (resampled PH) | |
| $\mathcal{T}_1 := \frac{1}{S} \sum_s \sum_{e \in \mathrm{MST}(X_c^{(s)})} w(e) \, \mathbf{1}[w(e) \geq \tau]$ | $m_0 = 800$, $\tau = 0.01$ |
| $\mathcal{T}_2 := \frac{1}{S} \sum_s \sum_\ell (d_\ell - b_\ell) \, \mathbf{1}[(d_\ell - b_\ell) \geq \tau]$ | $m_1 = 400$, $\tau = 0.01$ |
| $\mathcal{T}_3 := \frac{1}{S} \sum_s \sum_v (d_v - b_v) \, \mathbf{1}[(d_v - b_v) \geq \tau]$ | $m_2 = 384$, $L = 128$ |

**Separation subvector.** For a dataset $(\mathcal{D})$, we define the separation subvector

$$h_{\mathcal{SEP}}(\mathcal{D}) = \big[\mathcal{S}_1(\mathcal{D}), \ \mathcal{S}_2(\mathcal{D}), \ \mathcal{S}_3(\mathcal{D})\big].$$

**Separation failure modes.** "Lack of separation" can arise through qualitatively different mechanisms: local ambiguity ($\mathcal{S}_1$), hub-driven shortcuts ($\mathcal{S}_2$), or fragile inter-cluster margins ($\mathcal{S}_3$). (Radovanovic et al., 2010; Tomasev et al., 2014; Awasthi et al., 2012; Kumar & Kannan, 2010)

$\mathcal{S}_1$**: Neighbor overlap.** When clustering relies on neighborhood graphs, strong cross-label mixing in local neighborhoods weakens membership evidence and undermines graph-based recoverability assumptions (Von Luxburg, 2007). Local ambiguity is therefore measured as the fraction of $k$NN edges that cross reference labels.

$\mathcal{S}_2$**: Hubness infiltration.** Cross-label mixing is quantified by assessing whether it is disproportionately mediated by a small set of high-influence points (hubs) that appear in many $k$NN lists. Such hubness is a documented high-dimensional phenomenon that can distort neighborhood graphs and introduce shortcut-like connections, affecting downstream learning, including clustering (Radovanovic et al., 2010; Tomasev et al., 2014).

$\mathcal{S}_3$**: Normalized margin thickness.** Thin inter-cluster gaps relative to typical within-cluster neighbor spacing predict sensitivity to perturbations and limited tuning budgets: stability and perturbation-resilience analyses show that small metric perturbations can alter the optimum or hinder recovery when instances are not sufficiently stable, while prox-imity and separation conditions for center-based clustering require points to be decisively closer to their own center than to others (Awasthi et al., 2012; Bilu & Linial, 2012; Kumar & Kannan, 2010). The thickness of inter-cluster gaps is therefore measured relative to typical within-cluster neighbor spacing.

### 3.2.2. COHESION: SINGLE-SCALE REGULARITY VERSUS MULTI-SCALE HETEROGENEITY.

Separation alone does not guarantee recoverability: many clustering pipelines implicitly assume that clusters are internally "simple" under a *shared* notion of resolution—e.g., that a single bandwidth, neighborhood size, density threshold, or prototype scale is adequate across groups. This shared-scale assumption is violated by multi-scale density structure (core–halo clusters, heavy tails, and strong density imbalance) and by pronounced geometric anisotropy, both of which induce systematic merge–split trade-offs under global parameter choices. (Ester et al., 1996; Comaniciu & Meer, 2002; Fraley & Raftery, 2002; Durani et al., 2022)

**Cohesion subvector.** For a dataset $\mathcal{D}$, we define the cohesion subvector

$$h_{\mathcal{COH}}(\mathcal{D}) = \big[\mathcal{C}_1(\mathcal{D}), \ \mathcal{C}_2(\mathcal{D})\big],$$

which summarizes internal heterogeneity (scale/density) and anisotropy (shape).

**Cohesion profile: internal heterogeneity and anisotropy.** $\mathcal{C}_1$**: Multi-scale density heterogeneity.** Single-resolution clustering becomes brittle under heterogeneous densities: within-cluster multi-scale variation (e.g., core–halo structure or heavy low-density tails) and across-cluster density imbalance preclude a single global resolution. These effects are captured by measuring (i) within-cluster multi-scale density variation and (ii) across-cluster density imbalance (Ester et al., 1996; Comaniciu & Meer, 2002; Campello et al., 2015).

$\mathcal{C}_2$**: Linear elongation.** Filamentary, anisotropic cluster geometry induces failure modes distinct from overlap or density heterogeneity: elongated clusters encourage "chaining" under connectivity-based objectives and can lead to systematic over-splitting or incorrect merges when methods prefer roughly isotropic groups or rely on global neighborhoods (Sibson, 1973; Fraley & Raftery, 2002). Such geometry is quantified via the covariance spectrum, with large principal-component dominance captured by high $\mathcal{C}_2$.

### 3.2.3. TOPOLOGY AND CONNECTIVITY: MULTISCALE STRUCTURE BEYOND COMPACT BLOBS.

A complementary perspective defines clusters through *connectivity across scales*, particularly in density-based and

hierarchical settings where population structure is expressed as connected components of density upper level sets (the *cluster tree* view). (Hartigan, 1975; Chaudhuri & Dasgupta, 2010) Under this lens, difficulty can arise even with moderate separation when the data exhibits chain-like connectivity, narrow bridges that merge components early, or persistent non-contractible structure (loops/voids) that violates compact, convex, or center-based intuitions. (Chaudhuri & Dasgupta, 2010; Ghrist, 2008)

**Topology subvector.** For a dataset $\mathcal{D}$, we define the topology subvector

$$h_{\mathcal{TOPO}}(\mathcal{D}) = \big[\mathcal{T}_1(\mathcal{D}),\ \mathcal{T}_2(\mathcal{D}),\ \mathcal{T}_3(\mathcal{D})\big],$$

using scalable persistent-homology (PH) summaries. Persistent homology provides a standard multiscale summary of connectivity (PH0) and higher-order features such as loops (PH1) and voids (PH2). (Ghrist, 2008; Edelsbrunner & Harer, 2010; Edelsbrunner & Morozov, 2017)

**Topology profile (PH0–PH2).** $\mathcal{T}_1$**: PH0 component persistence.** Delayed merging of connected components across the filtration scale signals tenuous connectivity and chain or bridge structure: cutting too early fragments clusters, while narrow bridges induce premature merging in graph- and density-based procedures (Chaudhuri & Dasgupta, 2010; Ghrist, 2008). This behavior is summarized by PH0 persistence, with large values indicating delayed merging.

$\mathcal{T}_2$**: PH1 loop persistence.** Non-contractible structure, such as rings or holes, violates "single-center" or compact-blob intuitions and can induce clustering failures even when local overlap is low (Ghrist, 2008; Edelsbrunner & Harer, 2010). Such structure is summarized by the persistence of PH1 features.

$\mathcal{T}_3$**: PH2 void persistence.** Shell-like cavities and voids, which arise in scientific point clouds and learned embeddings, can confound methods that rely on density cores or compactness (Ghrist, 2008; Edelsbrunner & Harer, 2010). Such structure is summarized by the persistence of PH2 features.

### 3.3. CHB Evaluation

Aggregated ARI/NMI leaderboards obscure *where* methods fail and *why*. CHB adds algorithm-agnostic structural descriptors to each dataset–representation instance, enabling mechanism-level diagnosis. Our evaluation asks whether performance varies systematically along CHB axes, organized around three questions: **(Q1)** which structural regimes predict consistent success or failure; **(Q2)** how method families behave within regimes; and **(Q3)** how representation changes reshape structure—and thereby outcomes. We evaluate tabular datasets in native feature space and im-

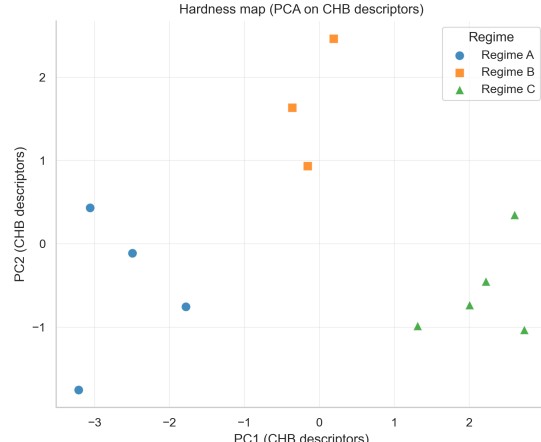

*Figure 2.* **Hardness map (descriptor-only PCA).** Each point is a base/raw dataset embedded by Principal-Component-Analysis using the 8 CHB descriptors. Colors indicate regimes A/B/C assigned by the rule in Section 4.

age datasets under multiple learned representations; these representation shifts serve as *structured interventions* whose movement in CHB space can be tested for association with clustering behavior. We benchmark diverse families (center-, density-, spectral-, hierarchical-, and deep methods, including topology-aware variants) under *default* and *budgeted tuning* tracks to separate structural mismatch from parameter sensitivity. Experiments span established image benchmarks (MNIST, KMNIST, FMNIST, CIFAR-10/100, STL-10, SVHN) and standard tabular datasets (Kelly et al., 2023). The concrete results can be found in our appendix, and we instead focus on the higher-level conclusions to be drawn from those results.

## 4. Regimes in CHB Space (Q1)

Clustering theory separates mechanism-level conditions for recovery. The most fundamental is *local identifiability/separation*; conditional on usable separation, additional regularities of *Topology* and *scale/heterogeneity* determine whether a method's inductive bias matches the data. Standard benchmarks, however, typically report only aggregated scores, which can obscure which structural mechanism is responsible for failure and when conclusions should transfer across datasets (Fig. 2).

CHB introduces an algorithm- and score-independent measurement layer that characterizes datasets through structure rather than outcomes. Given reference labels (which is standard in external benchmarking), CHB computes a *hardness fingerprint* without running clustering methods (Eq. 1).

**Regime construction (separability gate with blob-calibrated topology evidence).** We discretize the CHB

hardness fingerprint into three theory-aligned regimes using a separability gate, a blob-calibrated topology evidence statistic $\mathcal{T}_{\text{evid}}$, and a cohesion-heterogeneity severity axis $h_{\mathcal{TOPO}}$. Separability is evaluated first: if local neighborhoods are not membership-consistent, recoverability assumptions for both prototype- and graph-based objectives fail and performance degrades broadly. Conditional on separability, residual difficulty splits into topology/geometry mismatch (Regime B) versus scale and heterogeneity effects (Regime C); $h_{\mathcal{COH}}$ quantifies mismatch severity within each regime. **Separability gate.** Let $(\mathcal{S}_1, \mathcal{S}_2, \mathcal{S}_3)$ capture neighborhood mixing, hub-induced shortcuts, and normalized margin robustness. We declare separability failure when at least two indicators are in their strict failure regions:

$$\mathbb{I}[\mathcal{S}_1(\mathcal{D}) > \tau_1] + \mathbb{I}[\mathcal{S}_2(\mathcal{D}) > \tau_2] + \mathbb{I}[\mathcal{S}_3(\mathcal{D}) < \tau_3] \geq 2 \quad (2)$$

where $\mathbb{I}[\cdot]$ denotes the indicator function, and the default thresholds are $\tau_1 = 0.5$ (local-majority boundary), $\tau_2 = 0.33$ (hub-infiltration ceiling), and $\tau_3 = 1.0$ (unit-margin boundary). The gate *fails* (declaring separability collapse) when Eq. (2) holds; otherwise, the gate *passes* (Details in Appendix D.3). **Topology evidence with a blob reference.** For gate-passing datasets, we measure whether non-blob structure is present via

$$\mathcal{T}_{\text{evid}}(\mathcal{D}) = \log(1 + \mathcal{T}_1) + \log(1 + \mathcal{T}_2) + \log(1 + \mathcal{T}_3), \quad (3)$$

which aggregates multiscale connectivity ($\mathcal{T}_1$, PH0), loop persistence ($\mathcal{T}_2$, PH1), and void persistence ($\mathcal{T}_3$, PH2). Crucially, we calibrate "non-trivial topology" against an explicit Gaussian-blob null rather than suite-relative normalization: we generate isotropic Gaussian mixtures across several configurations spanning realistic $(n, d, k)$ ranges and compute $\mathcal{T}_{\text{evid}}$ with the identical pipeline. The resulting threshold $\tau_{\text{TOP}} = 15.0$ is the 95th percentile under this null, so $\mathcal{T}_{\text{evid}}(\mathcal{D}) > 15.0$ indicates topology unlikely to arise from blob-like clusters.

**Deterministic regime rule.** Regime membership is defined independently of any visualization by the fixed rule

$$r(\mathcal{D}) = \begin{cases} \textbf{A}, & \text{if the separability gate fails Eq. 2,} \\ \textbf{B}, & \text{if the gate passes and } \mathcal{T}_{\text{evid}}(\mathcal{D}) > 15.0, \\ \textbf{C}, & \text{if the gate passes and } \mathcal{T}_{\text{evid}}(\mathcal{D}) \leq 15.0. \end{cases}$$

This yields a clean semantic split: Regime A corresponds to separability collapse (local evidence is unreliable), whereas Regime B isolates post-gate difficulty driven primarily by Topology Mismatch versus Regime C scale/heterogeneity, respectively.

**Calibrated, theory-aligned thresholds.** Each gate threshold ($\mathcal{S}_1 > 0.5$, $\mathcal{S}_2 > 0.33$, $\mathcal{S}_3 < 1.0$) encodes a distinct, interpretable failure mode tied to clustering recoverability,

and the 2-of-3 logic enforces that the gate triggers only under systematic breakdown. The topology cutoff $\tau_{\text{TOP}} = 15.0$ is suite-independent by construction—anchored to a blob null—so Regime B is reserved for genuinely non-blob topological signals rather than artifacts of relative scaling. Finally, the descriptor-only Principal-Component-Analysis (PCA, Fig. 2) is used solely as an interpretability sanity check; it does not define regimes, which are fully determined by the rule above.

### 4.1. Regime semantics

We describe each regime following this hierarchy: first separation collapse (Regime A), then—among instances with usable separation—mismatch-driven difficulty (Regime B) versus scale-driven difficulty (Regime C). Descriptor magnitudes are described in Table 15.

**Regime A: Separability Collapse. Typical members:** CIFAR-10/100, STL-10, SVHN.
**Signature:** Strong local ambiguity—high neighborhood mixing and hub-mediated shortcutting ($\mathcal{S}_1, \mathcal{S}_2$ elevated) with only moderate margin robustness ($\mathcal{S}_3$); see Table 15.
**Mechanism and induced effects:** The separability gate fails: $k$NN neighborhoods are cross-class and hubs inject shortcut edges, so local membership evidence is unreliable. This failure typically propagates to global structure as *connectivity-scale inflation* (large $\mathcal{T}_1$): there is no stable intermediate scale at which class-aligned connectivity emerges. Elevated $\mathcal{T}_2/\mathcal{T}_3$ may co-occur, but often as downstream artifacts of the broken neighborhoods rather than genuine persistent non-trivial topology.
**Implication:** This regime is *representation-limited*: algorithm choice provides limited relief unless the representation rescues separability (reduce $\mathcal{S}_1/\mathcal{S}_2$ and improve $\mathcal{S}_3$; see Q3, Section 6).

**Regime B: Topology Mismatch. Typical members:** MNIST, KMNIST, FMNIST.
**Signature:** Usable local evidence (low-to-moderate $\mathcal{S}_1$ and $\mathcal{S}_2$ with stronger margins $\mathcal{S}_3$), but pronounced topological structure: persistent loops and voids (large $\mathcal{T}_2$ and $\mathcal{T}_3$), often with inflated connectivity scale ($\mathcal{T}_1$), and elevated anisotropy (high $\mathcal{C}_2$) (Table 15).
**Mechanism:** The separability gate passes: a *bootstrap signal* exists (neighborhoods are often class-consistent). However, clusters are not well-matched by compact "blob" assumptions. Anisotropy ($\mathcal{C}_2$) and manifold-like organization produce genuine non-trivial topology—persistent loops ($\mathcal{T}_2$) and voids ($\mathcal{T}_3$)—that induces systematic merge–split trade-offs for fixed-scale methods. Unlike Regime A, elevated $\mathcal{T}_2/\mathcal{T}_3$ here reflect *real* topological structure, not collapse artifacts.
**Implication:** This is a *recoverable-but-mismatched* regime. Gains come from mismatch correction—reducing effective

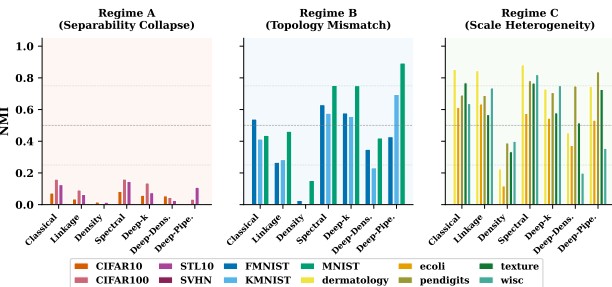

*Figure 3.* Regime-dependent ordering across algorithm families

anisotropy or topological entanglement—rather than creating separability from scratch.

**Regime C: Scale heterogeneity. Typical members:** Dermatology, Ecoli, Pendigits, Texture, Wisc (tabular).
**Signature:** Strong separability (low-to-moderate $\mathcal{S}_1$ and $\mathcal{S}_2$ with high $\mathcal{S}_3$), comparatively simple topology (low-to-moderate $\mathcal{T}_1$–$\mathcal{T}_3$), and residual difficulty dominated by non-exchangeable cohesion (moderate $\mathcal{C}_1$ with variable $\mathcal{C}_2$) (Table 15).
**Mechanism:** The separability gate passes with robust margins ($\mathcal{S}_3$ high). Topology is comparatively simple (low $\mathcal{T}$ indicators). Remaining difficulty stems from *non-exchangeable* cohesion: clusters differ in density, spread, or shape ($\mathcal{C}_1$, $\mathcal{C}_2$), so no single global choice of $k$, bandwidth, or density threshold fits all groups simultaneously.
**Implication:** The bottleneck is *scale selection under heterogeneity*. Methods robust to multi-scale structure or adaptive parameterization are favored (see Q2, Section 5).

# 5. Global Clusterability Regimes and Mechanism–Algorithm Coupling (Q2)

CHB regimes from Q1 correspond to a consistent **global ordering of recoverability** ($A < B < C$). Let $\tilde{p}_{R,m}$ denote method $m$'s median agreement over datasets in regime $R$, and define the regime-level cross-method summary $\tilde{p}_R := \text{median}_m \tilde{p}_{R,m}$ (Table 16). Empirically, agreement rises from $\tilde{p}_A \approx 0.067$ to $\tilde{p}_B \approx 0.529$ and peaks at $\tilde{p}_C \approx 0.704$. Beyond this global ordering, **which family wins depends on the active hardness mechanism**: Spectral methods lead in A and C (best method: Spectral), while deep-pipeline methods lead in B (best method: DDC). These regime-dependent shifts reflect coupling between algorithmic assumptions and CHB axes ($\mathcal{S}_1 - \mathcal{S}_3, \mathcal{C}_1 - \mathcal{C}_2, \mathcal{T}_1 - \mathcal{T}_3$): separability is most predictive of whether any family can succeed, while mismatch and heterogeneity correlate with which family is favored once separability is usable (full tables in Appendix E).

## 5.1. Regime-dependent ordering across algorithm families

**Family grouping.** We use a granular family taxonomy aligned with method assumptions: **classical-center** ($k$-means (Lloyd, 1982), GMM (Dempster et al., 1977)), **linkage** (Ward (Ward Jr, 1963)/average (Lance & Williams, 1967)), **Spectral** (Von Luxburg, 2007) (graph cuts; best understood as a classical-center method on a learned representation, i.e., Laplacian eigenmaps), **density** (DBSCAN (Ester et al., 1996)/HDBSCAN (Campello et al., 2013)), and three deep clustering subfamilies: **deep-$k$-center** (IDEC (Guo et al., 2017), DCN (Yang et al., 2017)), **deep-density** (SHADE (Beer et al., 2024)) , **deep-pipeline** (DDC (Ren et al., 2020)). For interpretability, we summarize each $(R, F)$ by the *best-performing method in family $F$* and its regime median (Table 17); dataset-level family medians are reported separately (Table 18, Fig. 3).

**Empirical family ordering is regime-dependent.**

**Regime A (separability collapse; near-universal failure).** All methods fail in absolute terms ($\tilde{p}_A \approx 0.067$). Spectral is highest but still weak ($\tilde{p}_{A,\text{Spectral}} \approx 0.115$; best method Spectral, Table 16), with classical-center and linkage close behind ($\approx 0.10$), while the remaining families are much lower (deep-$k$-center $\leq 0.084$, and deep-density/density/deep-pipeline $\leq 0.037$; Table 17). Descriptively, high local mixing and hubness ($\mathcal{S}_1, \mathcal{S}_2 \uparrow$) combined with insufficient separability ($\mathcal{S}_3$ weak) corrupt both neighborhood graphs and prototype objectives; deep self-training lacks a stable bootstrap signal, and density-based methods additionally break down in the absence of a coherent connectivity/density scale (consistent with the $\mathcal{T}_1 - \mathcal{T}_3$ signatures used in Q1).

**Regime B (mismatch; strongest method dependence).** Regime B exhibits a stronger separation among families ($\tilde{p}_B \approx 0.529$) than Regime A or C. deep-pipeline dominates ($\tilde{p}_{B,\text{deep-pipeline}} \approx 0.695$; best method DDC, Table 16), with Spectral as the closest competitor ($\approx 0.631$). A middle tier includes deep-$k$-center and linkage ($\approx 0.58$ and $\approx 0.57$), followed by classical-center ($\approx 0.49$), while deep-density is substantially lower ($\approx 0.35$) and density-based methods collapse ($\approx 0.05$; Table 17). Descriptively, separability is usable (higher $\mathcal{S}_3$, lower $\mathcal{S}_1/\mathcal{S}_2$), so pipeline self-training may reshape representations to reduce effective anisotropy and mismatch (elevated $\mathcal{C}_2$ and nontrivial $\mathcal{T}_1 - \mathcal{T}_3$), whereas single-scale density cues remain unreliable unless cohesion is stable.

**Regime C (heterogeneity; high but non-uniform).** Regime C is broadly recoverable ($\tilde{p}_C \approx 0.704$), but rankings reflect scale and cohesion heterogeneity. Spectral leads ($\tilde{p}_{C,\text{Spectral}} \approx 0.782$, Table 16), with linkage and deep-pipeline close behind ($\approx 0.728$ and $\approx 0.727$). Deep-$k$-

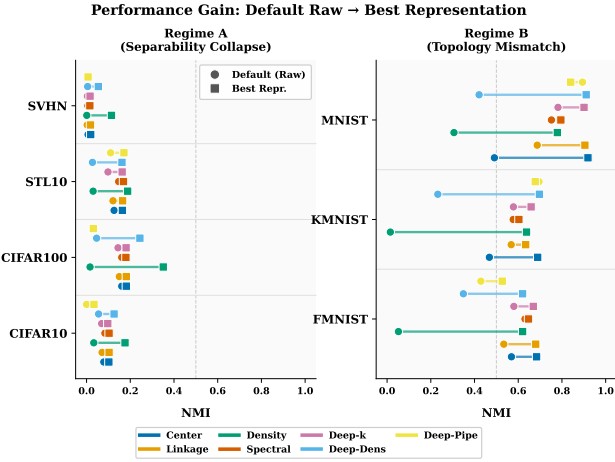

*Figure 4.* Representation learning shifts data features, thereby improving clustering performance. CHB explains where these gains come from.

center and classical-center form the next tier ($\approx 0.71$ and $\approx 0.70$), while density and deep-density remain far below the regime ceiling ($\approx 0.45$; Table 17). Mechanistically, once $\mathcal{S}_3$ is strong, many objectives work, but cross-cluster non-exchangeability and multi-scale cohesion (high $\mathcal{C}_1$ / heterogeneous neighborhoods) systematically penalize any method requiring a single global density/scale threshold, producing merge–split instability; graph-based and linkage-based approaches are comparatively stable under a coherent metric.

**Takeaway.** Q2 replaces global leaderboards with *mechanism-conditioned* rankings. Separability is the gate: when local evidence collapses (overlap, hubness, weak margins), all families fail similarly. Once separability is usable, performance is shaped by the violated assumptions: Topology Mismatch drives sharp, family-specific breakdowns and ranking reversals (notably for density- and graph-based methods), while cohesion heterogeneity systematically penalizes single-resolution procedures through merge–split instability.

# 6. Representation Auditing with CHB (Q3)

To study representation effects, we treat learned embeddings (e.g., UMAP (McInnes et al., 2018) and DDC outputs, details in Appendix F) as *structural interventions*: each dataset is first transformed in representation space and then re-embedded into the same CHB hardness space defined in Q1, without modifying the descriptors or the regime rule. This allows us to compare datasets directly in CHB space and to interpret representation gains as movement *within* or *between* regimes. CHB characterizes representation gains through two objects: **(i) regime transitions** in CHB space and **(ii) within-regime axis movement**. Empirically, large

dataset-level gains concentrate in B→C transitions, while A→A transitions yield smaller, often family-conditional improvements (Table 21, Fig. 4).

*Validation of the Q1 regime rule under representation shifts.* The Q1-based regime rule persists after representation learning: substantial gains arise primarily for instances that cross into the post-gate regime, while instances remaining gated (A→A) show little improvement.

This motivates a **separability gate**. In **Regime A** (separability collapse), broad gains are unlikely unless local neighborhood evidence becomes usable, i.e., overlap and hub-driven shortcutting decrease (typically $\mathcal{S}_1, \mathcal{S}_2 \downarrow$). In **Regime B** (separability usable), separability provides a reliable bootstrap, and improvements are often driven by **mismatch correction**: representations simplify topology (typically $\Delta \mathcal{T}_1 < 0$, $\Delta \mathcal{T}_2 < 0$, often $\Delta \mathcal{T}_3 < 0$, and sometimes $\Delta \mathcal{C}_2 < 0$) even when separability changes are modest.

Accordingly, CIFAR-10/CIFAR-100 (and similarly STL-10/SVHN) serve as negative controls: embeddings simplify mismatch without a consistent separability rescue, so improvements remain limited and family-conditional (Tables 21, 22–25). Conversely, MNIST/KMNIST/FMNIST show digit-family behavior where moving toward Regime C coincides with large gains and interpretable axis shifts (Tables 21, 23–28).

## 6.1. Auditing setup: representations, deltas, and regimes

**Hardness and performance deltas; Default vs. Best.** For a dataset $\mathcal{D}$, we denote its default representation also by $\mathcal{D}$, and let $\mathcal{D}_{emb}$ denote a learned representation of the same dataset. We define hardness changes $\Delta \mathbf{h}(\mathcal{D}; \mathcal{D}_{emb}) = \mathbf{h}(\mathcal{D}_{emb}) - \mathbf{h}(\mathcal{D})$. Let $p_m(\mathcal{D})$ denote the NMI of method $m$ and define the dataset score $\bar{p}(\mathcal{D}) = \max_{m \in \mathcal{M}} p_m(\mathcal{D})$. We select a single best learned representation $\mathcal{D}^*_{emb}(\mathcal{D}) \in \arg\max_{\mathcal{D}_{emb}} \bar{p}(\mathcal{D}_{emb})$ and report *Default* $\bar{p}(\mathcal{D})$, *Best* $\bar{p}(\mathcal{D}^*_{emb})$, and $\Delta \bar{p} = \bar{p}(\mathcal{D}^*_{emb}) - \bar{p}(\mathcal{D})$. Table 21 reports the corresponding *family-wise* scores, i.e., $\max_{m \in \mathcal{M}_F} p_m(\mathcal{D})$ and $\max_{m \in \mathcal{M}_F} p_m(\mathcal{D}^*_{emb})$ within each family $F$ for the default and best learned representations, respectively.

## 6.2. Two improvement modes under the separability gate

In our empirical study, CHB suggests two dominant modes, whose effect depends on the starting regime.

**Mode 1: separability rescue (unlocking local evidence).** A representation performs a rescue when it reduces overlap and hub shortcuts: $\Delta \mathcal{S}_1 < 0$, $\Delta \mathcal{S}_2 < 0$, often with $\Delta \mathcal{S}_3 > 0$. When this happens, broad gains become more plausible as neighborhood graphs and self-training signals become more reliable.

**Mode 2: mismatch correction (simplifying topology).** A representation performs mismatch correction when it primarily reduces topological complexity: $\Delta\mathcal{T}_1 < 0$, $\Delta\mathcal{T}_2 < 0$, and often $\Delta\mathcal{T}_3 < 0$, sometimes $\Delta\mathcal{C}_2 < 0$, with separability changes modest or mixed. This is associated with improved performance for classical and many deep objectives, while density methods remain sensitive to whether a stable operative scale/heterogeneity emerges (linked to $\mathcal{C}_1$ and margin/scale cues).

**Gate implication.** In **Regime A**, mismatch correction alone is often insufficient for *broad* gains; improvements (if any) are typically family-conditional (Table 21). In **Regime B**, separability already supports a bootstrap, so mismatch correction can translate into broad gains for center- and linkage-based methods (and often deep-density), while density methods remain sensitive to residual scale/heterogeneity and deep-pipeline methods may not improve (Tables 21, 20).

### 6.3. Why transitions are the main explanatory object

We interpret representation effects in two steps. First, regime transitions summarize whether the representation changed the *type* of difficulty. In our benchmark, best representations either keep datasets in Regime A (A→A) or move digit datasets from Regime B toward Regime C (B→C; Table 21). Second, axis deltas characterize *how* difficulty changed: within a fixed regime, gains should align with either separability rescue ($\mathcal{S}_1, \mathcal{S}_2 \downarrow$) or mismatch correction ($\mathcal{T}_1 - \mathcal{T}_3 \downarrow$, sometimes $\mathcal{C}_2 \downarrow$).

### 6.4. Empirical audits: negative controls and the digit trilogy

**Negative controls (A→A): CIFAR-10/100, STL-10, SVHN.** These datasets remain in Regime A under best raw-feature representation variants (UMAP/DDC variants) (Table 21). Mismatch indicators drop substantially ($\mathcal{T}_1, \mathcal{T}_2 \downarrow$; $\mathcal{T}_3$ often decreasing), yet separability shows no consistent rescue—$\mathcal{S}_1$ and $\mathcal{S}_2$ remain elevated or increase, while $\mathcal{S}_3$ is mixed (Tables 22–25). Performance follows the gate: gains are modest and family-concentrated. Notably, density-based methods improve sharply on CIFAR-10/100 ($0.033 \rightarrow 0.176$; $0.016 \rightarrow 0.351$) despite persistent separability collapse—representation shifts repair local connectivity ($\mathcal{C}_1$), benefiting density objectives even without broader rescue. **Positive cases (B→C): MNIST, KMNIST, FMNIST.** These datasets exhibit large, broad gains under best representations (Table 21) with interpretable axis shifts: **MNIST/KMNIST:** Strong rescue signature ($\mathcal{S}_1, \mathcal{S}_2 \downarrow$; $\mathcal{S}_3 \uparrow$) plus topology simplification ($\mathcal{T}_1$–$\mathcal{T}_3 \downarrow$), yielding competitive performance across multiple families (Tables 28–27). Notably, the spectral embedding follows the same B→C trajectory as UMAP and DDC, confirming that CHB's diagnostics generalise across embedding families. **FMNIST:**

Mismatch-dominant pattern—topology simplifies strongly while separability shifts are smaller and mixed. Gains are substantial but narrower; density methods do not uniformly dominate (Table 26).

### 6.5. Stress test: modern representation pipeline

To test whether CHB generalizes beyond raw features, we track regime trajectories through a CLIP (Radford et al., 2021) → UMAP pipeline on image and text datasets (Table 29). The separability gate at the intermediate CLIP stage already predicts final outcomes: CIFAR-10 passes the gate after CLIP (A→B→C, $\Delta$NMI $= +.713$), while SVHN remains gated throughout (A→A, $\Delta$NMI $= -.044$). The same pattern holds across modalities—AG News (Zhang et al., 2015) reaches Regime C with large gains, while 20 Newsgroups stays in A with marginal improvement (details in Appendix F.1).

**Takeaway.** Representation learning corresponds to movement in CHB space, and improvement modes follow the separability-first semantics (Q1). When embeddings reduce mismatch without improving neighborhood evidence (A→A), gains are real but typically family-conditional (Tables 22–25, 21). When embeddings rescue separability and reduce mismatch (B→C; digits), improvements are large and often broad across families (Tables 26–28, 21). *Separability rescue enables broad gains; mismatch correction predicts which families benefit once separability holds.*

## 7. Conclusion

CHB frames clustering evaluation as *hardness-aware diagnosis*. By embedding datasets through separability, cohesion, and topology, it reveals the mechanisms behind success and failure. Empirically, separability is the main gate to recoverability. Once this gate is passed, topology mismatch drives family-specific reversals, while cohesion heterogeneity degrades single-resolution methods. Representation learning follows the same structure, with movement in CHB space predicting whether gains come from restoring separability or correcting family-dependent mismatch.

## 8. Limitations and Future Work

CHB currently targets metric point-cloud data, covering most clustering benchmarks. Future work will extend CHB to graph-structured data through shortest-path separation, edge-weight filtrations, and graph-native cohesion. CHB also assumes reference labels only for diagnosis, so unsupervised proxies such as stability-based separability surrogates could broaden use in unlabeled settings. Finally, while PH0–PH2 suffices for the regimes observed here, PH3+ features may help on richer geometries at higher computational cost.

## Impact Statement

This paper presents work whose goal is to advance the field of Machine Learning. There are many potential societal consequences of our work, none of which we feel must be specifically highlighted here.

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

| Appendix Section | Content |
|---|---|
| Appendix A | CHB framework |
| Appendix B | CHB space |
| Appendix C | Experimental Setup and Evaluation Protocol |
| Appendix D | Q1: Regime Construction and Empirical Validation |
| Appendix E | Q2: Family-Specific Failure Modes |
| Appendix F | Q3: Representation shift |
| Appendix G | Ablation Study: Separation, Cohesion, Topology |
| Appendix H | Hyperparameter Tuning in CHB's Space |
| Appendix I | Stability Analysis |
| Appendix J | Internal cluster metrics and CHB space |

*Table 4.* Structure of the appendix.

# A. CHB framework

In Fig. 5 we outline the full CHB framework.

# CHB Framework: Hardness-Aware Clustering Evaluation

*From aggregate leaderboards to mechanism-level diagnosis*

**Input:** $(\mathcal{D}, y)$ – dataset + reference labels
*(labels for diagnosis only; never used to fit clustering)*

**Output:** Regime $\in \{A, B, C\}$ + fingerprint $\mathbf{h}(\mathcal{D})$
+ regime-conditioned performance

**The Hardness Fingerprint $\mathbf{h}(\mathcal{D}) = (\mathcal{S}; \mathcal{C}; \mathcal{T})$**

*Descriptor-only: no clustering runs required*

**Separation ($\mathcal{S}$)** – *Local identifiability*

| | | |
|---|---|---|
| $\mathcal{S}_1$ Overlap | Cross-label $k$NN mixing ($\uparrow$ worse) |
| $\mathcal{S}_2$ Hubness | Hub-mediated shortcuts ($\uparrow$ worse) |
| $\mathcal{S}_3$ Margins | Normalized gap thickness ($\uparrow$ better) |

**Cohesion ($\mathcal{C}$)** – *Scale compatibility*

| | | |
|---|---|---|
| $\mathcal{C}_1$ Density | Multi-scale heterogeneity ($\uparrow$ harder) |
| $\mathcal{C}_2$ Shape | Linear elongation ($\uparrow$ harder) |

**Topology ($\mathcal{T}$)** – *Structural complexity*

*(scalable: subsampled + MST proxy + radius-capped)*

| | | |
|---|---|---|
| $\mathcal{T}_1$ PH$_0$ | Component persistence ($\uparrow$ delayed merge) |
| $\mathcal{T}_2$ PH$_1$ | Loop persistence ($\uparrow$ non-convex) |
| $\mathcal{T}_3$ PH$_2$ | Void persistence ($\uparrow$ shell-like) |

**Regime Assignment Rule** *(semantic cutoffs + null-calibrated topology)*

**Separability Gate** (2-of-3 failure triggers collapse)

Gate FAILS if:
$$\mathbb{I}[S_1(\mathcal{D}) > \tau_1] + \mathbb{I}[\mathcal{S}_2(\mathcal{D}) > \tau_2] + \mathbb{I}[\mathcal{S}_3(\mathcal{D}) < \tau_3] \geq 2$$

**Topology Evidence** ($\tau_{\text{TOP}} = 15.0$: 95% under Gaussian-blob null)

$$T_{\text{evid}}(\mathcal{D}) = \log(1+\mathcal{T}_1) + \log(1+\mathcal{T}_2) + \log(1+\mathcal{T}_3)$$

**Regime Assignment**

| | | |
|---|---|---|
| **A** | Gate FAILS | Separability Collapse |
| **B** | Pass $\wedge$ $\mathcal{T}_{\text{evid}} > 15$ | Topology Mismatch |
| **C** | Pass $\wedge$ $\mathcal{T}_{\text{evid}} \leq 15$ | Scale heterogeneity |

## Key Empirical Findings (Q1–Q3)

**Q1: Regime Structure**

| | |
|---|---|
| A: | CIFAR-10/100, STL-10, SVHN |
| B: | MNIST, KMNIST, FMNIST |
| C: | Pendigits, Texture, Wisc |

**Q2: Family$\times$Regime Coupling**

| | A | B | C |
|---|---|---|---|
| Median NMI | 0.07 | 0.53 | 0.70 |
| Best family | Spec. | Deep-P. | Spec. |

**Q3: Representation Lens**

Repr. learning $\Rightarrow$ movement in CHB space

A$\rightarrow$A: $C\downarrow, T\downarrow$ but $S$ not rescued
  $\Rightarrow$ bounded, family-conditional gains
B$\rightarrow$C: $S$ rescued + $T\downarrow$
  $\Rightarrow$ broad gains (MNIST: +40% NMI)

*Core insight:* Geometry shifts ($C\downarrow, T\downarrow$) **cannot escape the separation gate** without rescuing $S \Rightarrow$ leaderboards mislead without regime conditioning

**Workflow:** $\boxed{(\mathcal{D}, y)} \xrightarrow{\text{compute}} \boxed{\mathbf{h}(\mathcal{D})} \xrightarrow{\text{gate}} \boxed{\text{Regime}} \xrightarrow{\text{condition}} \boxed{\text{Diagnosis}} \xrightarrow{\text{attribute}} \boxed{\mathcal{S} \text{ vs } \mathcal{C} \text{ vs } \mathcal{T}}$

*Figure 5.* CHB framework overview. **Labels are for diagnosis only; never used to fit clustering.** The fingerprint $\mathbf{h}(\mathcal{D})$ is descriptor-only and captures three failure mechanisms. Separation thresholds are semantic; $\tau_{\text{TOP}}$ **is null-calibrated, making it suite-independent.** Key insight: representation learning shifts geometry but cannot escape the separation gate without rescuing $\mathcal{S}$.

## B. CHB space

CHB maps each dataset $\mathcal{D}$ to an interpretable *hardness fingerprint* $\mathbf{h}(\mathcal{D})$. The fingerprint concatenates three descriptor blocks:

$$\mathbf{h}(\mathcal{D}) = \big[\, h_{\mathcal{SEP}}(\mathcal{D})\,;\, h_{\mathcal{COH}}(\mathcal{D})\,;\, h_{\mathcal{TOPO}}(\mathcal{D})\,\big].$$

Intuitively, $h_{\mathcal{SEP}}$ captures how reliably local neighborhoods support cluster membership, $h_{\mathcal{COH}}$ captures scale/shape mismatch inside clusters, and $h_{\mathcal{TOPO}}$ summarizes multiscale connectivity and higher-order geometric structure.

**Setup and notation.**   All descriptors are computed *label-conditionally* with respect to a reference partition $y \in \{1, \ldots, K\}^n$. Let $X_r = \{x_i\}_{i=1}^n \subset \mathbb{R}^d$ be the representation, and let $C_c := \{i : y_i = c\}$ denote the index set of cluster $c$. We also write $X_c := \{x_i : i \in C_c\}$ and $\bar{C}_c := \{1, \ldots, n\} \setminus C_c$.

**Robust dataset-level aggregation (used throughout).**   Each descriptor is first computed per cluster, yielding values $\{v_c\}_{c=1}^K$. We then aggregate to a dataset-level scalar using a cluster-size-weighted 10% trimmed mean:

$$\mathrm{Agg}(\{v_c\}_{c=1}^K) := \mathrm{TrimMean}_{0.10}\Big(\{v_c\}; \text{ weights } w_c = |C_c|/n\Big).$$

Several descriptors are evaluated over small deterministic internal grids (e.g., neighborhood sizes); to reduce sensitivity to any single scale, we aggregate *within* each cluster by a median over the grid before applying $\mathrm{Agg}$ across clusters.

### B.1. Separation: local identifiability and robustness

Separation measures whether local neighborhoods provide consistent membership evidence and whether inter-cluster gaps are robust. We define:

$$h_{\mathcal{SEP}}(\mathcal{D}) = \big[\mathcal{S}_1(\mathcal{D}),\, \mathcal{S}_2(\mathcal{D}),\, \mathcal{S}_3(\mathcal{D})\big].$$

The three coordinates progressively move from (i) local outgoing neighborhood ambiguity, to (ii) incoming-neighborhood "shortcut" contamination, to (iii) robust inter-cluster margins.

$\mathcal{S}_1$**: Neighbor overlap (local ambiguity).**   Let $\mathcal{N}_k(i)$ be the indices of the $k$ nearest neighbors of $x_i$ (excluding $i$) (Cover & Hart, 1967). Define the local cross-label fraction:

$$O_k(i) := \frac{1}{k} \sum_{j \in \mathcal{N}_k(i)} \mathbb{I}[y_j \neq y_i].$$

For cluster $c$, define the mean at scale $k$:

$$\mu_c^{(k)} := \frac{1}{|C_c|} \sum_{i \in C_c} O_k(i).$$

To stabilize across scales, aggregate over a small internal grid $\mathcal{K}_{\mathcal{S}_1}$ by median:

$$\mu_c := \mathrm{median}_{k \in \mathcal{K}_{\mathcal{S}_1}} \mu_c^{(k)}.$$

Finally, aggregate across clusters:

$$\boxed{\mathcal{S}_1 := \mathrm{Agg}(\{\mu_c\}_{c=1}^K).} \tag{4}$$

*Orientation:* higher $\mathcal{S}_1$ indicates stronger local label mixing (worse separability).

*Complexity:* Dominated by $k$-NN construction for $k_{\max} := \max \mathcal{K}_{\mathcal{S}_1}$. With exact pairwise distances: $\mathcal{O}(n^2 d)$. With spatial indexing/ANN backends in moderate dimension: $\mathcal{O}(nd \log n + n k_{\max})$. Aggregating across the grid is $\mathcal{O}(n|\mathcal{K}_{\mathcal{S}_1}|)$.

$$\boxed{\mathcal{O}(n^2 d) \text{ (exact)} \quad \text{or} \quad \mathcal{O}(nd \log n + n k_{\max}) \text{ (indexed/ANN)}}$$

$\mathcal{S}_2$: **Hubness infiltration (shortcut contamination).** While $\mathcal{S}_1$ looks at a point's *outgoing* neighborhood, $\mathcal{S}_2$ measures whether points become cross-cluster "hubs" in *incoming* $k$NN relations (Radovanovic et al., 2010). For a fixed $k$, define the in-degree (hubness) of point $v$ in the directed $k$NN graph:

$$h_k(v) := \big|\{u :\ v \in \mathcal{N}_k(u)\}\big|.$$

Define the cross-label in-degree:

$$h_k^{\mathrm{cross}}(v) := \big|\{u :\ v \in \mathcal{N}_k(u)\ \wedge\ y_u \neq y_v\}\big|.$$

The per-point infiltration ratio is:

$$I_k(v) := \begin{cases} h_k^{\mathrm{cross}}(v)/h_k(v), & h_k(v) > 0, \\ 0, & h_k(v) = 0. \end{cases}$$

For cluster $c$, compute the mean infiltration at scale $k$:

$$\mu_c^{(k)} := \frac{1}{|C_c|} \sum_{v \in C_c} I_k(v).$$

Aggregate across an internal grid $\mathcal{K}_{\mathcal{S}_2}$ and then across clusters:

$$\mu_c := \mathrm{median}_{k \in \mathcal{K}_{\mathcal{S}_2}}\ \mu_c^{(k)}.$$

$$\boxed{\mathcal{S}_2 := \mathrm{Agg}\big(\{\mu_c\}_{c=1}^K\big).} \tag{5}$$

*Orientation:* higher $\mathcal{S}_2$ indicates stronger hub-mediated cross-cluster shortcuts (worse separability).

*Complexity:* Reuses the $k$-NN graph (up to $k_{\max} := \max \mathcal{K}_{\mathcal{S}_2}$). In-degree counts can be computed in $\mathcal{O}(nk_{\max})$ per $k$ (or $\mathcal{O}(nk_{\max})$ total with a prefix-count implementation). Overall:

$$\boxed{\mathcal{O}(n^2 d)\ \text{(exact)} \quad \text{or} \quad \mathcal{O}(nd \log n + nk_{\max})\ \text{(indexed/ANN)}}$$

$\mathcal{S}_3$: **Normalized margin thickness.** The first two coordinates quantify ambiguity and contamination. In contrast, $\mathcal{S}_3$ measures the *thickness of inter-cluster gaps* relative to the cluster's internal spacing (Awasthi et al., 2012; Bilu & Linial, 2012; Kumar & Kannan, 2010).

*Within-cluster scale.* For cluster $c$, let $r_{k_0}^{(c)}(i)$ be the distance from $x_i$ to its $k_0$-th nearest neighbor *within* $C_c$ (excluding self). Over a small grid $\mathcal{K}_0$, define:

$$s_c := \mathrm{median}_{k_0 \in \mathcal{K}_0}\ \mathrm{median}_{i \in C_c}\ r_{k_0}^{(c)}(i).$$

*kNN outside-margin.* For each $i \in C_c$, let $\delta_p^{(c)}(i)$ be the distance from $x_i$ to its $p$-th nearest neighbor in $\bar{C}_c$. For a fixed quantile $q \in (0,1)$:

$$m_c^{\mathrm{kNN}}(p) := \mathrm{Quantile}_q\big(\{\delta_p^{(c)}(i)\}_{i \in C_c}\big), \qquad \rho_c^{\mathrm{kNN}}(p) := \frac{m_c^{\mathrm{kNN}}(p)}{s_c}.$$

*Aggregation.* Aggregate over a small grid $\mathcal{P}$ and then across clusters:

$$\rho_c := \mathrm{median}_{p \in \mathcal{P}}\ \rho_c^{\mathrm{kNN}}(p).$$

$$\boxed{\mathcal{S}_3 := \mathrm{Agg}(\{\rho_c\}_{c=1}^K).} \tag{6}$$

*Orientation:* higher $\mathcal{S}_3$ indicates thicker, more robust inter-cluster gaps (better).

*Complexity:* Within-cluster $k$-NN queries cost $\mathcal{O}(\sum_c |C_c|^2 d)$ exactly or $\mathcal{O}(\sum_c |C_c| d \log |C_c|)$ with indexing; cross-cluster outside-margin queries cost $\mathcal{O}(n^2 d)$ exactly or $\mathcal{O}(n\, d \log n)$ with indexing (for fixed $p$).

$$\boxed{\mathcal{O}(n^2 d)\ \text{(exact)} \quad \text{or} \quad \mathcal{O}(n\, d \log n)\ \text{(indexed/ANN)}}$$

## B.2. Cohesion: single-scale regularity vs. multi-scale heterogeneity

Cohesion summarizes whether clusters admit a compatible internal "resolution" and whether their shape deviates from compact, roughly isotropic blobs. We define:

$$h_{\mathcal{COH}}(\mathcal{D}) = \big[\mathcal{C}_1(\mathcal{D}),\ \mathcal{C}_2(\mathcal{D})\big].$$

$\mathcal{C}_1$**: Multi-scale density heterogeneity (Density-Complexity).**   $\mathcal{C}_1$ captures two effects: (i) within-cluster density spread and (ii) cross-cluster density imbalance, both evaluated across multiple neighborhood scales.

For each $k$ in an internal grid $\mathcal{K}_D$, define a normalized $k$NN density proxy. Let $\bar{r}_k(i)$ be the mean distance from $x_i$ to its $k$ nearest neighbors (excluding self) and set $\tilde{d}_k(i) := 1/(\bar{r}_k(i) + \epsilon)$. Implementation applies a robust transform: (i) replace non-finite values by the maximum finite density, (ii) clip at the 99th percentile, and (iii) min–max normalize to $[0, 1]$. Denote the resulting value by $d_k(i) \in [0, 1]$ (Loftsgaarden & Quesenberry, 1965).

For cluster $c$, define spread and median at scale $k$:

$$\text{spread}_{c,k} := Q_{0.90}\big(\{d_k(i) : i \in C_c\}\big) - Q_{0.10}\big(\{d_k(i) : i \in C_c\}\big), \qquad \text{med}_{c,k} := Q_{0.50}\big(\{d_k(i) : i \in C_c\}\big).$$

Let $\text{med}_{\text{glob},k} := \text{median}_c\ \text{med}_{c,k}$ be the global median across clusters at scale $k$, and define a one-sided between-cluster penalty:

$$\text{between}_{c,k} := \max\{\text{med}_{\text{glob},k} - \text{med}_{c,k},\ 0\}.$$

Per-scale density complexity is then:

$$\text{DC}_{c,k} := \text{spread}_{c,k} + \text{between}_{c,k}.$$

Aggregate within cluster across scales and then across clusters:

$$C1_c := \text{median}_{k \in \mathcal{K}_D}\ \text{DC}_{c,k}.$$

$$\boxed{\mathcal{C}_1 := \text{Agg}(\{C1_c\}_{c=1}^{K}).} \tag{7}$$

*Orientation:* higher $\mathcal{C}_1$ indicates stronger multi-scale heterogeneity (worse for single-resolution methods).

*Complexity:* Dominated by $k$NN queries for density estimation.

$$\boxed{\mathcal{O}(n^2 d)\ \text{(exact)} \quad \text{or} \quad \mathcal{O}(nd \log n + nk_{\max})\ \text{(indexed/ANN)}, \ \ k_{\max} := \max \mathcal{K}_D}$$

$\mathcal{C}_2$**: Linear elongation (anisotropy/filamentarity).**   Let $\{\lambda_j^{(c)}\}_{j=1}^{r_c}$ be the positive eigenvalues of the centered cluster scatter, sorted decreasingly (Abdi & Williams, 2010). Define:

$$C2_c := \begin{cases} 0, & r_c \leq 1, \\[2mm] \dfrac{\lambda_1^{(c)}}{\frac{1}{r_c-1}\sum_{j=2}^{r_c}\lambda_j^{(c)}}, & r_c \geq 2. \end{cases}$$

$$\boxed{\mathcal{C}_2 := \text{Agg}(\{C2_c\}_{c=1}^{K}).} \tag{8}$$

*Orientation:* higher $\mathcal{C}_2$ indicates stronger anisotropy/filamentarity (higher mismatch risk for blob-like assumptions).

*Complexity:* Summing thin SVD costs over clusters:

$$\boxed{\mathcal{O}\Big(\sum_{c=1}^{K} |C_c| \cdot d^2\Big) = \mathcal{O}(nd^2)}$$

Note: This is independent of $k$-NN and does not require Ripser.

## B.3. Topology: multiscale structure beyond compact blobs

Topology summarizes multiscale connectivity (PH0) and higher-order features (PH1/PH2) using scalable, resampled persistence summaries with a lifetime threshold $\tau > 0$. We define:

$$h_{\mathcal{TOPO}}(\mathcal{D}) = \big[ \mathcal{T}_1(\mathcal{D}),\ \mathcal{T}_2(\mathcal{D}),\ \mathcal{T}_3(\mathcal{D}) \big].$$

All three descriptors follow the same pattern: subsample within each cluster, compute a persistence proxy on each subsample, threshold short-lived features, then average and aggregate. In the following $S$ (number of resamples per cluster) $= 1$ for all PH metrics.

$\mathcal{T}_1$**: PH0 component persistence (MST-based, resampled).** For each cluster $c$, draw $S$ subsamples $X_c^{(s)} \subseteq X_c$ of size $\min\{|C_c|, m_0\}$. Compute an MST on $X_c^{(s)}$ (exact on the complete graph when feasible; otherwise via an MST on a connected $k$NN graph (Gower & Ross, 1969)). Let $w(e)$ denote MST edge weights. Define the PH0 proxy on subsample $s$:

$$T1_c^{(s)} := \sum_{e \in \mathrm{MST}(X_c^{(s)})} w(e) \cdot \mathbb{I}[w(e) \geq \tau].$$

Average over subsamples and aggregate across clusters:

$$T1_c := \frac{1}{S} \sum_{s=1}^{S} T1_c^{(s)}.$$

$$\boxed{\mathcal{T}_1 := \mathrm{Agg}(\{T1_c\}_{c=1}^{K}).} \tag{9}$$

*Orientation:* higher $\mathcal{T}_1$ indicates delayed merging / inflated connectivity scale (harder).

*Complexity:* Per cluster, per subsample of size $m_0$:

- **Exact MST:** pairwise distances $\mathcal{O}(m_0^2 d)$, then Prim/Kruskal $\mathcal{O}(m_0^2 \log m_0)$.

- **Approximate MST ($k$NN graph):** $k$NN $\mathcal{O}(m_0 d \log m_0)$, sparse MST $\mathcal{O}(m_0 k \log(m_0 k))$.

Total over all clusters and subsamples:

$$\boxed{\mathcal{O}\big(K \cdot S \cdot m_0^2 \cdot (d + \log m_0)\big)}$$

$\mathcal{T}_2$**: PH1 loop persistence (resampled, radius-capped).** For each subsample $X_c^{(s)}$ (size $\min\{|C_c|, m_1\}$), compute a Vietoris–Rips filtration, optionally capped at radius $R$ (maximum edge length) (Edelsbrunner & Harer, 2010). Let $\{(b_\ell, d_\ell)\}_\ell$ be PH1 bars with finite death. Define:

$$T2_c^{(s)} := \sum_\ell (d_\ell - b_\ell) \cdot \mathbb{I}[(d_\ell - b_\ell) \geq \tau], \qquad T2_c := \frac{1}{S} \sum_{s=1}^{S} T2_c^{(s)}.$$

$$\boxed{\mathcal{T}_2 := \mathrm{Agg}(\{T2_c\}_{c=1}^{K}).} \tag{10}$$

*Orientation:* higher $\mathcal{T}_2$ indicates more persistent loop structure (harder).

*Complexity:* For $m_1$ points, the VR complex has $\mathcal{O}(m_1^2)$ edges and up to $\mathcal{O}(m_1^3)$ triangles, though radius capping reduces this substantially in practice. We summarize as:

$$\boxed{\mathcal{O}\big(K \cdot S \cdot m_1^2 \cdot \alpha(R, \mathrm{topology})\big)}$$

$\mathcal{T}_3$**: PH2 void persistence (resampled, landmarked, radius-capped).** Analogously, compute PH2 on subsamples of size $\min\{|C_c|, m_2\}$, optionally using landmark VR when subsamples are large, and optionally capping the filtration radius by $R$. Let $\{(b_v, d_v)\}_v$ be PH2 bars with finite death (De Silva & Carlsson, 2004). Define:

$$T3_c^{(s)} := \sum_v (d_v - b_v) \cdot \mathbb{I}[(d_v - b_v) \geq \tau], \qquad T3_c := \frac{1}{S} \sum_{s=1}^{S} T3_c^{(s)}.$$

$$\boxed{\mathcal{T}_3 := \mathrm{Agg}(\{T3_c\}_{c=1}^K).} \tag{11}$$

*Orientation:* higher $\mathcal{T}_3$ indicates more persistent void/shell structure (harder).

*Complexity:* PH2 introduces 3-simplices; worst-case $\mathcal{O}(m_2^4)$, mitigated by landmarks ($L$) and radius capping:

$$\boxed{\mathcal{O}\big(K \cdot S \cdot \min(m_2^3,\ Lm_2^2) \cdot \beta(R, \text{topology})\big)}$$

**Summary of the fingerprint and orientations.** CHB yields:

$$\mathbf{h}(\mathcal{D}) = \big[\,(\mathcal{S}_1, \mathcal{S}_2, \mathcal{S}_3)\,;\,(\mathcal{C}_1, \mathcal{C}_2)\,;\,(\mathcal{T}_1, \mathcal{T}_2, \mathcal{T}_3)\,\big].$$

We interpret each coordinate directionally as a mechanism-strength indicator Table 7:

- $\mathcal{S}_1$ (neighbor overlap): higher $\Rightarrow$ worse separability (more cross-label $k$NN mixing).
- $\mathcal{S}_2$ (hubness infiltration): higher $\Rightarrow$ worse separability (hub-mediated shortcut contamination).
- $\mathcal{S}_3$ (normalized margin thickness): higher $\Rightarrow$ better separability/robustness (thicker gaps relative to within-cluster scale).
- $\mathcal{C}_1$ (DensityComplexity): higher $\Rightarrow$ worse single-resolution fit (global $k$/bandwidth/threshold becomes brittle).
- $\mathcal{C}_2$ (linear elongation): higher $\Rightarrow$ stronger anisotropy/filamentarity (merge–split trade-offs, chaining risk).
- $\mathcal{T}_1$ (PH0 persistence): higher $\Rightarrow$ more chain/bridge connectivity and inflated connectivity scale.
- $\mathcal{T}_2$ (PH1 persistence): higher $\Rightarrow$ more persistent loop structure (non-blob geometry).
- $\mathcal{T}_3$ (PH2 persistence): higher $\Rightarrow$ more persistent void/shell structure (can confound compactness heuristics).

Parameter robustness analyses are detailed in Appendix I.

## B.4. Computational complexity summary

Table 5 summarizes the computational complexity of each CHB metric, where $n$ denotes the number of samples, $d$ the dimensionality, $K$ the number of clusters, and $m_i$ the subsample sizes for topology descriptors. For $k$NN-heavy terms, the table reports the indexed/ANN regime; exact brute-force substitutes $nd \log n \mapsto n^2 d$.

**Overall pipeline complexity.** Assuming the $k$-NN graph is computed once and shared across $\mathcal{S}_1$, $\mathcal{S}_2$, and $\mathcal{C}_1$:

$$\underbrace{\mathcal{O}(nd \log n)}_{k\text{NN backbone}} + \underbrace{\mathcal{O}(nd^2)}_{\mathcal{C}_2} + \underbrace{\mathcal{O}\big(K(m_0^2 d + m_1^2 \alpha + m_2^3 \beta)\big)}_{\text{Topology}}.$$

For typical benchmark datasets, the $k$NN construction dominates; topology remains tractable due to fixed $m_i \ll n$ and radius capping.

*Table 5.* Computational complexity of CHB metrics (indexed/ANN $k$NN regime). Exact brute-force substitutes $nd \log n \mapsto n^2 d$. topology metrics use subsampling ($m_i \ll n$) and radius capping to control cost.

| Block | Metric | Complexity | Bottleneck |
|---|---|---|---|
| | $\mathcal{S}_1$ | $\mathcal{O}(nd \log n + nk_{\max})$ | $k$-NN graph construction |
| Separation | $\mathcal{S}_2$ | $\mathcal{O}(nd \log n + nk_{\max})$ | Shared $k$-NN + in-degree counts |
| | $\mathcal{S}_3$ | $\mathcal{O}(n\,d \log n)$ | Within-cluster + cross-cluster $k$NN queries |
| Cohesion | $\mathcal{C}_1$ | $\mathcal{O}(nd \log n + nk_{\max})$ | $k$-NN density estimation |
| | $\mathcal{C}_2$ | $\mathcal{O}(nd^2)$ | SVD per cluster |
| | $\mathcal{T}_1$ | $\mathcal{O}(KSm_0^2 d)$ | Pairwise distances + MST |
| Topology | $\mathcal{T}_2$ | $\mathcal{O}(KSm_1^2 \alpha)$ | Ripser (PH1, radius-capped) |
| | $\mathcal{T}_3$ | $\mathcal{O}(KS \min(m_2^3, Lm_2^2)\beta)$ | Ripser (PH2, landmarked) |

*Table 6.* CHB metrics with final notation, default hyperparameters, and computational complexity (indexed/ANN $k$NN regime).

| Block | Metric | Default definition / values | Complexity |
|---|---|---|---|
| **Separation** | | | |
| | $\mathcal{S}_1$ | Neighbor overlap: cross-label $k$NN fraction, $\mathcal{K}_{\mathcal{S}_1} = \{10, 20, 40\}$ | $\mathcal{O}(nd \log n + n \cdot 40)$ |
| | $\mathcal{S}_2$ | Hubness infiltration: cross-label incoming $k$NN, $\mathcal{K}_{\mathcal{S}_2} = \{10, 20, 40\}$ | $\mathcal{O}(nd \log n + n \cdot 40)$ |
| | $\mathcal{S}_3$ | Normalized margin: $q = 0.25$, $\mathcal{P} = \{3\}$ | $\mathcal{O}(n\,d \log n)$ |
| **Cohesion / Scale** | | | |
| | $\mathcal{C}_1$ | Density complexity: multi-scale $k$NN spread, $\mathcal{K}_D = \{10, 15, 20\}$ | $\mathcal{O}(nd \log n + n \cdot 20)$ |
| | $\mathcal{C}_2$ | Linear elongation: $\lambda_1 / \mathrm{mean}(\lambda_{2:})$ | $\mathcal{O}(nd^2)$ |
| **Topology (Persistent Homology, resampled)** | | | |
| | $\mathcal{T}_1$ | PH0 (MST-based): $m_0 = 800$, $S = 1$, $\tau = 0.01$ | $\mathcal{O}(KSm_0^2 d)$ |
| | $\mathcal{T}_2$ | PH1 (loops): $m_1 = 400$, $S = 1$, $\tau = 0.01$ | $\mathcal{O}(KSm_1^2 \alpha)$ |
| | $\mathcal{T}_3$ | PH2 (voids): $m_2 = 384$, $L = 128$, $S = 1$, $\tau = 0.01$ | $\mathcal{O}(KSm_2^3 \beta)$ |
| **Aggregation (all metrics)** | | | |
| | – | Per-cluster $\rightarrow$ cluster-size-weighted 10% trimmed mean | $\mathcal{O}(K \log K)$ |

*Table 7.* CHB Metric Ranges and Interpretations. Theoretical ranges are based on metric definitions; empirical ranges are derived from Table 15.

| Block | Metric | Theoretical Range | Empirical Range | Orientation | Threshold (Gate) |
|---|---|---|---|---|---|
| | $\mathcal{S}_1$ (Overlap) | $[0, 1]$ | $[0.01, 0.91]$ | $\uparrow$ worse | $> \tau_1 = 0.5$ |
| **Separation** | $\mathcal{S}_2$ (Hubness) | $[0, 1]$ | $[0.02, 0.49]$ | $\uparrow$ worse | $> \tau_2 = 0.33$ |
| | $\mathcal{S}_3$ (Margin) | $[0, \infty)$ | $[0.71, 2.10]$ | $\uparrow$ better | $< \tau_3 = 1.0$ |
| **Cohesion** | $\mathcal{C}_1$ (Density) | $[0, 1]$ | $[0.04, 0.56]$ | $\uparrow$ harder | — |
| | $\mathcal{C}_2$ (Elongation) | $[1, \infty)$ | $[2.5, 4208]$ | $\uparrow$ harder | — |
| | $\mathcal{T}_1$ (PH0) | $[0, \infty)$ | $[49, 68692]$ | $\uparrow$ harder | |
| **Topology** | $\mathcal{T}_2$ (PH1) | $[0, \infty)$ | $[2.1, 222]$ | $\uparrow$ harder | $\mathcal{T}_{\mathrm{evid}} > 15.0$ |
| | $\mathcal{T}_3$ (PH2) | $[0, \infty)$ | $[0.16, 24.5]$ | $\uparrow$ harder | |

# C. Experimental details

We now outline the experimental details.
Code: https://github.com/Walid10010/CHB.git

## C.1. Dataset and further insights

Tables 8 and 9 present a comprehensive summary of the dataset statistics.

*Table 8.* Overview of all image datasets

| Dataset | Samples | Clusters | Dimensions |
|---|---|---|---|
| MNIST (LeCun et al., 1998) | 60000 | 10 | 784 |
| STL-10 (Coates et al., 2011) | 5000 | 10 | 27648 |
| CIFAR-100 (Krizhevsky & Hinton, 2009) | 50000 | 100 | 3072 |
| SVHN (Netzer et al., 2011) | 73257 | 10 | 3072 |
| FMNIST (Xiao et al., 2017) | 60000 | 10 | 784 |
| KMNIST (Clanuwat et al., 2018) | 60000 | 10 | 784 |
| CIFAR-10 (Krizhevsky & Hinton, 2009) | 50000 | 10 | 3072 |

*Table 9.* Stats for tabular datasets

| Dataset | Samples | Clusters | Dimensions |
|---|---|---|---|
| Pendigits (Kelly et al., 2023) | 10992 | 10 | 16 |
| Dermatology (Kelly et al., 2023) | 358 | 6 | 34 |
| Ecoli (Kelly et al., 2023) | 336 | 8 | 7 |
| Wisc (Kelly et al., 2023) | 699 | 2 | 9 |
| optdigits (Kelly et al., 2023) | 5620 | 10 | 62 |
| Texture (Kelly et al., 2023) | 5500 | 11 | 40 |

## C.2. Experimental Setup and Evaluation Protocol

This appendix specifies the experimental protocol executed by our batch runner for all CHB clustering evaluations. The description matches the released Python implementation (see Appendix listing) and is intended to be fully reproducible.

**Preprocessing.** All feature dimensions are scaled to $[0, 1]$ using MinMaxScaler. The number of target clusters is defined by the number of unique ground-truth labels,

$$K = |\text{unique}(y)|.$$

Ground-truth labels are never used to fit models. They are used only to compute evaluation metrics and, in the budgeted tuning track, to select the best configuration according to the reported metric.

**Evaluated methods.** We evaluate a consistent set of classical and deep clustering baselines:

- **Classical:** K-means, GMM, Spectral, DBSCAN, HDBSCAN, AggWard, AggAverage.

- **Deep:** DCN, IDEC, DDC, and the topology-aware method SHADE, implemented via the clustpy.deep library.

**Clustering Methods and Taxonomy** Clustering methods aim to partition data such that similar data points are grouped together while dissimilar ones are separated. However, clustering approaches differ substantially in how similarity is defined and how groups are formed, ranging from fuzzy and hierarchical assignments to strict hard partitions. In this work, we focus exclusively on *hard partitioning* methods, as they yield a unique cluster assignment for each data point and thus allow for straightforward and consistent comparison across approaches.

We organize the considered methods using a granular family taxonomy aligned with their underlying assumptions. Specifically, we distinguish four traditional clustering families and three deep clustering subfamilies.

*Table 10.* Overview of evaluated clustering methods.

| Family | Methods |
|---|---|
| Classical | K-means, GMM, Spectral, DBSCAN, AggWard, AggAverage |
| Deep | DCN, IDEC, DDC, SHADE |

*Table 11.* Reported metric and special label handling.

| Item | Definition | Notes |
|---|---|---|
| Primary metric | $\text{NMI}(\hat{y}, y)$ | reported for every run |
| DBSCAN noise | label $-1$ | included as separate label in NMI |

Among traditional approaches, **classical-center** methods represent clusters via explicit centers and assign data points based on proximity, exemplified by $k$-means (Lloyd, 1982) and Gaussian mixture models (GMM) (Dempster et al., 1977). **linkage** methods build clusters through hierarchical merging based on pairwise distances, for which we consider Ward linkage (Ward Jr, 1963) and average linkage (Lance & Williams, 1967). **Spectral** methods rely on graph-based representations and graph cut objectives to capture potentially non-convex cluster structures (Von Luxburg, 2007). Finally, **density** methods define clusters as connected regions of high sample density; we include DBSCAN (Ester et al., 1996) and its hierarchical extension HDBSCAN (Campello et al., 2013).

Deep clustering methods incorporate learned representations into the clustering process. Within this category, **deep-$k$-center** methods combine deep embeddings with center-based clustering objectives, as exemplified by IDEC (Guo et al., 2017) and DCN (Yang et al., 2017). **deep-density** methods target density-connected structures in learned embedding spaces, represented here by SHADE (Beer et al., 2024). Lastly, **deep-pipeline** approaches follow a sequential strategy in which deep feature extraction is combined with subsequent clustering stages; we use DDC (Ren et al., 2020) as a representative of this family.

**Representation learning.** For image datasets, we evaluate multiple representation variants, including UMAP embeddings at different target dimensions, and the latent representations produced by DDC. Each representation is treated as a distinct dataset instance and is analyzed using the same default CHB metric configuration. This isolates the effect of representation choice on separability, cohesion, and topology without conflating it with descriptor tuning. By comparing CHB fingerprints across variants of the same dataset, we attribute performance changes to measurable structural shifts—such as separability rescue or topology simplification—rather than to algorithmic or hyperparameter effects. Suffixes such as UMAP64, UMAP4, and DDC denote alternative representations of the same dataset. The suffix indicates either the target embedding dimensionality (e.g., UMAP64 = 64-dimensional UMAP embedding) or a learned latent representation (DDC). Each variant is treated as a separate dataset instance and analyzed using identical CHB settings; only the representation changes.

**Evaluation metric and label handling.** For every run, we report Normalized Mutual Information (NMI) between predicted assignments $\hat{y}$ and ground truth $y$. For DBSCAN, noise points (label $-1$) are treated as an additional cluster label and included in the metric computation.

### C.3. Evaluation Tracks

All methods are evaluated under identical tracks with explicit budgets.

#### C.3.1. (T1) DEFAULT TRACK

Each method is run once using a default configuration Tab. 12.

#### C.3.2. (T2) BUDGETED TUNING TRACK (LIGHT RANDOM SEARCH)

To simulate limited-budget hyperparameter optimization, each method is evaluated for LIGHT_EVALS= 25 randomly sampled configurations from predefined ranges. The configuration with the highest NMI is retained (Tab. 13).

*Table 12.* Default Track (T1): default configurations.

| Method | Default configuration |
|---|---|
| K-means | $n\_init = 10$, $max\_iter = 300$, $init =$ K-means++ |
| GMM | $covariance\_type =$ full, $reg\_covar = 10^{-6}$, $n\_init = 1$ |
| Spectral(NN affinity) | $n\_neighbors = 10$, $assign\_labels =$ K-means |
| DBSCAN | $eps = 0.5$, $min\_samples = 5$, $n\_jobs = -1$ |
| HDBSCAN | $min\_samples = 5$, $n\_jobs = -1$ |
| AggWard | Ward linkage, Euclidean metric |
| AggAverage | Average linkage, Euclidean metric |
| DCN | $pretrain\_epochs = 10$, $clustering\_epochs = 20$ |
| IDEC | $pretrain\_epochs = 10$, $clustering\_epochs = 20$ |
| DDC | $pretrain\_epochs = 10$ |
| SHADE | $n\_neighbors = 10$ |

*Table 13.* Budgeted Tuning Track (T2): random-search spaces.

| Method | Search space |
|---|---|
| K-means | $n\_init \sim \mathrm{Unif}\{5, 20\}$; $max\_iter \sim \mathrm{Unif}\{100, 500\}$; $init \in \{$ K-means++, random$\}$ |
| GMM | $covariance\_type \in \{$ full, diag$\}$; $reg\_covar \sim \log \mathrm{Unif}[10^{-6}, 10^{-2}]$; $n\_init \sim \mathrm{Unif}\{1, 5\}$ |
| Spectral(NN) | $n\_neighbors \sim \mathrm{Unif}\{5, 30\}$; $assign\_labels \in \{$ K-means, discretize$\}$ |
| DBSCAN/ | $eps \sim \mathrm{Unif}[0.1, 3.0]$; $min\_samples \sim \mathrm{Unif}\{3, 20\}$ |
| AggWard / AggAverage | no tunable hyperparameters |
| DCN / IDEC | $pretrain\_epochs \sim \mathrm{Unif}\{5, 20\}$; $clustering\_epochs \sim \mathrm{Unif}\{10, 40\}$ |
| DDC | $pretrain\_epochs \sim \mathrm{Unif}\{5, 20\}$ |
| SHADE | no tunable hyperparameters; single evaluation |

# D. Q1: Regime Construction and Empirical Validation

This appendix provides complete details on regime construction, including theory-motivated mechanisms, empirical validation, and supporting analyses.

## D.1. Motivation and Principles

**Goal.**   The objective of Q1 is to construct a *performance-free* taxonomy of clustering problem structure. Rather than defining difficulty using method scores—which would bake algorithmic biases into the taxonomy—we characterize each dataset instance by its CHB descriptor vector and validate that the resulting organization is aligned with theory-motivated failure mechanisms. This provides a stable scaffold for CHB: Q2 conditions method comparisons on regime, and Q3 interprets representation gains as movement within or between regimes.

## D.2. Dataset Pool and Descriptor Space

**Dataset pool (base/raw inputs).**   Q1 regime construction uses the *base* representation for each dataset: raw tabular features for tabular datasets and raw pixel space for image datasets. The base/raw pool comprises 12 datasets spanning three modalities:

- **Natural images:** CIFAR-10, CIFAR-100, STL-10, SVHN

- **Digit-like images:** MNIST, KMNIST, FMNIST

- **Tabular:** Dermatology, Ecoli, Pendigits, Texture, Wisc

**Descriptor space.**   For each dataset $D$ under its base representation, we compute an 8-dimensional CHB fingerprint:

$$\mathbf{h}(\mathcal{D}) = (\mathcal{S}_1, \mathcal{S}_2, \mathcal{S}_3, \mathcal{C}_1, \mathcal{C}_2, \mathcal{T}_1, \mathcal{T}_2, \mathcal{T}_3),$$

where:

- $\mathcal{S}_1$ (neighbor overlap): local cross-label mixing

- $\mathcal{S}_2$ (hubness infiltration): hub-mediated shortcut contamination

- $\mathcal{S}_3$ (margin thickness): inter-cluster gap robustness

- $\mathcal{C}_1$ (density complexity): multi-scale density heterogeneity

- $\mathcal{C}_2$ (elongation): linear anisotropy/filamentarity

- $\mathcal{T}_1$ (PH0 persistence): component connectivity scale

- $\mathcal{T}_2$ (PH1 persistence): loop/cycle structure

- $\mathcal{T}_3$ (PH2 persistence): void/cavity structure

**Key principle: theory motivates, data validates.**   Clustering theory motivates a small set of mechanism-distinct failure modes (e.g., local identifiability, structural alignment, and scale compatibility), but it does not prescribe a fixed number of regimes in real benchmark suites. We therefore adopt a parsimonious three-way taxonomy aligned with these mechanisms and validate empirically that it is stable and interpretable in CHB descriptor space.

## D.3. Separability Gate

**Scope and intent.**   This section provides a principled interpretation of the separability gate used by CHB. Our goal is *diagnostic rigor*: we anchor interpretable cutoffs for the three separation descriptors and re-express the original strict "2-of-3" gate as a single robust scalar score that is *exactly equivalent* to the original binary decision. We do *not* claim a new clusterability theorem; rather, we justify why these cutoffs encode minimal, interpretable local-evidence conditions and why robust aggregation is appropriate for regime construction.

Separability descriptors and minimal local evidence

Let $X = \{x_i\}_{i=1}^n$ be the representation of dataset $D$ under a feature map $r$ and let $y_i \in \{1, \ldots, K\}$ be reference labels used for external evaluation. Let $N_k(i)$ be the $k$ nearest neighbors of $x_i$ (excluding $i$), and consider the directed $k$NN graph with edges $(i \rightarrow j)$ iff $j \in N_k(i)$.

CHB summarizes separability using three complementary quantities: (i) $\mathcal{S}_1$ (neighborhood label mixing; outgoing ambiguity), (ii) $\mathcal{S}_2$ (hub-mediated shortcut contamination; incoming influence), and (iii) $\mathcal{S}_3$ (normalized margin thickness; gap vs. within-cluster scale). These capture distinct failure modes of local identifiability and robustness, and are aggregated robustly across clusters and a small internal scale grid (median over scales, then a trimmed mean over clusters).

## D.4. Anchoring the three cutoffs

### D.4.1. $\mathcal{S}_1$: THE LOCAL MAJORITY BOUNDARY AT $\tau_1 = 0.5$

Define the per-point outgoing cross-label fraction at scale $k$:

$$O_k(i) := \frac{1}{k} \sum_{j \in N_k(i)} \mathbf{1}[y_j \neq y_i], \qquad A_k(i) := 1 - O_k(i) = \mathbb{P}(y_J = y_i \mid J \sim \mathrm{Unif}(N_k(i))).$$

CHB aggregates $O_k(i)$ by first averaging within each reference cluster, taking a median across a small $k$-grid, and then aggregating across clusters (robustly). Thus, $\mathcal{S}_1$ can be read as a robust dataset-level summary of typical outgoing cross-label mixing.

**Proposition D.1** (Local majority evidence and the $\tau_1 = 0.5$ boundary (diagnostic))**.** *The cutoff $\mathcal{S}_1 = \tau_1 = 0.5$ corresponds to a* local majority boundary*: values $\mathcal{S}_1 > 0.5$ indicate that typical local neighborhoods are foreign-majority (outgoing mixing exceeds $1/2$), undermining a minimal bootstrap signal from neighborhood structure.*

**Interpretation.** The cutoff $\tau_1 = 0.5$ is semantic rather than tuned: it is the unique threshold separating same-label majority from foreign-label majority in local neighborhoods. This aligns with the use of neighborhood structure in $k$NN graphs, Spectral affinities, and density connectivity, and with the separability-first hierarchy motivating CHB's regime construction.

### D.4.2. $\mathcal{S}_2$: INTERPRETING $\tau_2 = 0.33$ VIA WITHIN-VS-CROSS INFLUENCE

CHB's $\mathcal{S}_2$ measures *incoming* shortcut contamination in the directed $k$NN graph. For node $v$, let $h_k(v)$ be its in-degree and let $h_k^{\mathrm{cross}}(v)$ be the number of incoming edges originating from different labels. Define the infiltration ratio

$$I_k(v) := \begin{cases} h_k^{\mathrm{cross}}(v)/h_k(v), & h_k(v) > 0, \\ 0, & h_k(v) = 0. \end{cases}$$

CHB aggregates $I_k(v)$ across a small $k$-grid and across clusters (robustly) to obtain $\mathcal{S}_2$.

**A semantic yardstick.** A hub is problematic when its cross-label influence is not marginal relative to within-label influence, creating shortcut-like edges that can distort $k$NN-based affinity structure. One interpretable way to read a cutoff near $1/3$ is as a *within-label supermajority* requirement for influential nodes:

$$h_k^{\mathrm{same}}(v) \geq 2\, h_k^{\mathrm{cross}}(v) \quad \Longleftrightarrow \quad I_k(v) \leq \frac{1}{3}.$$

CHB uses $\tau_2 = 0.33$, close to $1/3$.

### D.4.3. $\mathcal{S}_3$: NORMALIZED MARGIN BOUNDARY AT $\tau_3 = 1$

CHB's $\mathcal{S}_3$ measures separation thickness relative to within-cluster scale. The boundary $\mathcal{S}_3 = \tau_3 = 1$ is semantic:

$$\mathcal{S}_3 > 1 \quad \text{means "gap thicker than scale," while} \quad \mathcal{S}_3 \leq 1 \quad \text{means "gap no thicker than scale."}$$

When $\mathcal{S}_3 \leq 1$, small metric perturbations (or small representation shifts) can plausibly flip nearest-competitor relations; we therefore treat $\mathcal{S}_3 > 1$ as a measurable robustness proxy rather than a formal guarantee.

**Note on directionality.** $\mathcal{S}_3$ is oriented oppositely to $\mathcal{S}_1$ and $\mathcal{S}_2$: higher $\mathcal{S}_3$ indicates *better* separability. Consequently, the failure condition for $\mathcal{S}_3$ is $\mathcal{S}_3 < \tau_3$, whereas for $\mathcal{S}_1$ and $\mathcal{S}_2$ it is $> \tau_j$.

## D.5. Median-of-failure-margins: a robust aggregation exactly equivalent to CHB 2-of-3

CHB operationalizes systematic separability collapse via a strict "2-of-3" gate: the gate fails only when at least two of the three indicators lie in their strict failure regions.

**Failure margins.** Define three *failure margins* (positive = strict failure, non-positive = not-failure):

$$f_1 := \mathcal{S}_1 - \tau_1, \qquad f_2 := \mathcal{S}_2 - \tau_2, \qquad f_3 := \tau_3 - \mathcal{S}_3,$$

with default thresholds $\tau_1 = 0.5$, $\tau_2 = 0.33$, and $\tau_3 = 1.0$.

**Definition D.2** (Separability failure severity score). Define the continuous (piecewise-linear) failure severity score

$$\mathrm{SEPF}(\mathcal{D}) := \mathrm{median}\{f_1, f_2, f_3\}.$$

**Proposition D.3** (Exact equivalence to the strict 2-of-3 separability gate). *The following are equivalent:*

1. $\mathrm{SEPF}(\mathcal{D}) > 0$;

2. *at least two of the three strict failure conditions hold:*

$$\mathcal{S}_1 > \tau_1, \qquad \mathcal{S}_2 > \tau_2, \qquad \mathcal{S}_3 < \tau_3;$$

3. *equivalently, the separability gate fails iff* $\mathrm{SEPF}(\mathcal{D}) > 0$, *and passes otherwise.*

*Proof.* For three real numbers, the median is positive iff at least two numbers are positive. Applying this to $\{f_1, f_2, f_3\}$ yields the claim. $\square$

**Note.** The median-of-three is a canonical robust aggregator: a single marginal violation cannot flip the gate unless at least one other descriptor agrees. This formalizes the intuition behind "2-of-3" without any independence assumption and provides a scalar $\mathrm{SEPF}(\mathcal{D})$ that can be plotted and used in sensitivity analyses while preserving the original binary gate.

## D.6. Final gate definition

**Separability gate.** We implement the separability gate via the equivalent strict median-of-failure-margins rule:

$$\boxed{\mathrm{GateFail}(\mathcal{D}) \iff \mathrm{SEPF}(\mathcal{D}) = \mathrm{median}\{\mathcal{S}_1 - \tau_1, \ \mathcal{S}_2 - \tau_2, \ \tau_3 - \mathcal{S}_3\} > 0,} \tag{12}$$

with default thresholds $\tau_1 = 0.5$, $\tau_2 = 0.33$, and $\tau_3 = 1.0$.

**Regime assignment.** After passing the separability gate, we retain the blob-calibrated topology evidence split used in CHB: instances with strong Topology Mismatch evidence form Regime B, and those with blob-like topology form Regime C. This preserves the separability-first hierarchy (A: separability collapse; B/C: separability-usable with topology split).

**Topology evidence (blob-referenced).** For the post-gate topology test, we anchor the threshold to a *Gaussian-blob null reference*. We define the topology evidence statistic:

$$\mathcal{T}_{\mathrm{evid}}(\mathcal{D}) := \log(1 + \mathcal{T}_1) + \log(1 + \mathcal{T}_2) + \log(1 + \mathcal{T}_3),$$

which aggregates connectivity ($\mathcal{T}_1$), loop ($\mathcal{T}_2$), and void ($\mathcal{T}_3$) persistence into a single scalar. The threshold $\tau_{\mathrm{TOP}}$ is set to the 95th percentile of $\mathcal{T}_{\mathrm{evid}}(\mathcal{D})$ under a Gaussian-blob null, computed across 147 valid blob configurations spanning $n \in \{1000, \ldots, 100000\}$, $d \in \{5, 10, 20, 50\}$, and $k \in \{2, \ldots, 100\}$:

$$\tau_{\mathrm{TOP}} = 15.0$$

This model-based threshold answers: *how large can topology appear when clusters are maximally blob-like?* Values exceeding $\tau_{\mathrm{TOP}}$ indicate genuine geometric complexity unlikely to arise from isotropic Gaussian structure ($< 5\%$ probability under the null).

**Cohesion score (mechanistic severity).** The cohesion block $(\mathcal{C}_1, \mathcal{C}_2)$ quantifies density heterogeneity and cluster elongation. We apply a log-transform to the heavy-tailed elongation descriptor:

$$\mathcal{C}_2^{\log} := \log(1 + \mathcal{C}_2)$$

Cohesion does *not* define the B/C boundary; instead it provides a mechanistic explanation of difficulty within Regime C, where scale heterogeneity rather than topological mismatch drives clustering challenges.

**Deterministic regime assignment.** Regimes are assigned by a separability gate followed by a blob-referenced topology test:

$$r(\mathcal{D}) = \begin{cases} \mathbf{A}, & \text{if separability gate fails (Eq. 2),} \\ \mathbf{B}, & \text{if gate passes and } \mathcal{T}_{\text{evid}}(\mathcal{D}) > \tau_{\text{TOP}}, \\ \mathbf{C}, & \text{if gate passes and } \mathcal{T}_{\text{evid}}(\mathcal{D}) \leq \tau_{\text{TOP}}. \end{cases} \tag{13}$$

### D.7. CHB Threshold Gate

Both thresholds have principled, theory-grounded justifications:

- **Separability gate**: Each threshold $(\mathcal{S}_1 > \tau_1, \mathcal{S}_2 > \tau_2, \mathcal{S}_3 < \tau_3)$ corresponds to a meaningful failure mode—majority foreign neighbors, substantial hub shortcuts, or sub-unit margins. The "2-of-3" rule provides robustness against single marginal violations.

- **Topology threshold**: $\tau_{\text{TOP}} = 15.0$ is the 95th percentile of topology evidence under a Gaussian-blob null model, providing a suite-independent reference for "more complex than blobs."

**Frozen calibration for representation auditing (Q3).** For Q3, we embed every dataset–representation instance using the same calibration constants $(\mu_x, \sigma_x)$ and the same thresholds $(\tau_1, \tau_2, \tau_3, \tau_{\text{TOP}})$ learned from the Q1 base/raw pool. This defines a single shared CHB coordinate system in which representation effects are interpreted as movement in CHB space rather than artifacts of re-standardization.

### D.8. Empirical validation on the base/raw pool

**Base/raw regime membership.** Applying Eq. (13) to the 12 base/raw datasets recovers the intended modality-aligned organization:

- **Regime A (separability collapse):** CIFAR-10, CIFAR-100, STL-10, SVHN.

- **Regime B (usable separability, topology/mismatch salient):** MNIST, KMNIST, FMNIST.

- **Regime C (usable separability, topology simple):** Dermatology, Ecoli, Pendigits, Texture, Wisc.

This organization matches the qualitative regime semantics encoded by the descriptor signatures in Table 15, and ensures that datasets described as "near-trivial topology" remain in Regime C under the post-gate rule.

**Stability.** The base/raw pool exhibits clear separation along the gate axis (separability) and along the post-gate topology axis ($\mathcal{T}_{\text{evid}}(\mathcal{D})$), so regime membership is stable under small perturbations of thresholds and under alternative monotone scalings. In particular, the post-gate decision depends on whether topology is meaningfully present (above baseline), not on a fragile comparison between two weak signals.

### D.9. Robustness to separability-gate thresholds

The separability gate (Eq. 2) uses interpretable default thresholds $\tau_1 = 0.5$, $\tau_2 = 0.33$, and $\tau_3 = 1.0$ to detect systematic local-identifiability breakdown. Its output is consumed by the deterministic regime rule (Eq. 13). To verify that regime membership is not a knife-edge artifact of these numeric cutoffs, we perform a targeted sensitivity analysis that perturbs only the gate thresholds while keeping the descriptor computation pipeline and the blob-referenced topology cutoff $\tau_{\text{TOP}}$ fixed, thereby isolating threshold effects from measurement or calibration changes.

*Table 14.* Final dataset selection used in the regime-based analysis

| Dataset | Regime | Remarks |
|---|---|---|
| CIFAR-10 | A | High overlap and hubness; |
| CIFAR-100 | A | Extreme overlap and hubness; inflated connectivity scale |
| SVHN | A | High overlap; weak margins; strong anisotropy |
| STL-10 | A | High overlap; degenerate connectivity structure |
| FMNIST | B | Moderate overlap; strong margins; nontrivial topology |
| KMNIST | B | Low–moderate overlap; strong margins; topology-driven mismatch |
| MNIST | B | Similar to KMNIST; separability with topological tension |
| Dermatology | C | Very strong separability; minimal topology |
| Ecoli | C | Cohesive clusters; weak topological signal |
| Pendigits | C | Extreme separability; minimal PH signal |
| Texture | C | Strong separability; density heterogeneity |
| Wisc | C | Very strong separability; near-trivial topology |

**Perturbation protocol.** Let $m \in \{0.8, 1.2\}$ denote a $\pm 20\%$ multiplicative perturbation. For each threshold $\tau_j$ with $j \in \{1, 2, 3\}$, we form a one-at-a-time perturbed gate by replacing $\tau_j$ with $\tau_j^{(m)} = m \cdot \tau_j$ while leaving the other two thresholds at their default values. The perturbed gate failure condition is:

$$\mathrm{GateFails}^{(m)}(\mathcal{D}) \iff \mathbb{I}[\mathcal{S}_1(\mathcal{D}) > \tau_1^{(m)}] + \mathbb{I}[\mathcal{S}_2(\mathcal{D}) > \tau_2^{(m)}] + \mathbb{I}[\mathcal{S}_3(\mathcal{D}) < \tau_3^{(m)}] \geq 2, \tag{14}$$

where only one of $\tau_1^{(m)}, \tau_2^{(m)}, \tau_3^{(m)}$ differs from its baseline value in any given run.

**Directionality.** The perturbation preserves the original gate semantics: decreasing $\tau_1$ or $\tau_2$ makes the gate *stricter* (more likely to fail), while *increasing* $\tau_3$ makes it stricter (since failure requires $\mathcal{S}_3 < \tau_3$). Note that $\mathcal{S}_3$ is oriented oppositely to $\mathcal{S}_1$ and $\mathcal{S}_2$: higher $\mathcal{S}_3$ indicates better separability. Although $\tau_1 = 0.5$ has the semantic interpretation of a "majority foreign neighbors" boundary, the $\pm 20\%$ sweep serves strictly as a robustness stress test probing both stricter and more lenient gating.

**Results.** We summarize sensitivity using agreement of the gate decision and final regime label relative to the baseline rule. On the Q1 base/raw pool (Table 15), regime membership is unchanged for all one-at-a-time $\pm 20\%$ perturbations of $\tau_1$, $\tau_2$, and $\tau_3$. Membership also remains unchanged under simultaneous $\pm 20\%$ perturbations of all three thresholds. This stability is consistent with the intended "2-of-3" logic, which suppresses dependence on any single marginal violation and triggers Regime A only under systematic separability collapse. The post-gate B/C split remains anchored by the fixed blob-referenced topology test ($\tau_{\mathrm{TOP}} = 15.0$) rather than by fragile threshold comparisons.

**D.10. Supporting visualization via PCA**

We use PCA as a compact visualization of standardized descriptor space and as a sanity check that dominant variation aligns with the intended mechanisms. PCA is *not* used as a decision rule for regime membership.

**Regime geometry in PC1–PC2 space (visual summary).** Figure 2 displays the base/raw datasets in PC1–PC2 space. Three well-separated regions are visible: Regime A occupies the separation-collapse region (high overlap/hubness and weak margins), while Regime B and Regime C lie in the separability-usable region and separate primarily by the mismatch axis (topology vs. heterogeneity). These regions summarize the descriptor-defined structure; regime membership is defined by Eq. (13) in the full standardized descriptor space and interpreted via the signatures in Table 15.

*Table 15.* CHB space for different dataset.

| dataset | $\mathcal{S}_1$ | $\mathcal{S}_2$ | $\mathcal{S}_3$ | $\mathcal{C}_1$ | $\mathcal{C}_2$ | $\mathcal{T}_1$ | $\mathcal{T}_2$ | $\mathcal{T}_3$ |
|---|---|---|---|---|---|---|---|---|
| CIFAR-100 | 0.913 | 0.494 | 0.712 | 0.413 | 213.508 | 19872.247 | 153.531 | 14.042 |
| CIFAR-10 | 0.706 | 0.363 | 0.827 | 0.404 | 1050.048 | 32842.853 | 112.766 | 7.108 |
| FMNIST | 0.183 | 0.155 | 0.971 | 0.378 | 237.775 | 11827.791 | 136.288 | 16.803 |
| KMNIST | 0.083 | 0.098 | 1.053 | 0.426 | 107.019 | 17216.209 | 180.687 | 22.851 |
| MNIST | 0.081 | 0.107 | 1.096 | 0.196 | 67.236 | 12102.879 | 179.404 | 24.541 |
| STL-10 | 0.752 | 0.344 | 0.792 | 0.443 | 103.878 | 68691.631 | 221.926 | 10.840 |
| SVHN | 0.634 | 0.419 | 0.770 | 0.559 | 4208.126 | 23871.255 | 182.303 | 11.701 |
| Dermatology | 0.056 | 0.039 | 1.261 | 0.383 | 4.012 | 173.094 | 6.862 | 0.795 |
| Ecoli | 0.173 | 0.192 | 1.062 | 0.372 | 2.514 | 49.371 | 2.056 | 0.163 |
| Pendigits | 0.011 | 0.018 | 2.096 | 0.451 | 12.403 | 561.048 | 20.376 | 1.974 |
| Texture | 0.029 | 0.032 | 1.470 | 0.332 | 115.373 | 410.841 | 21.006 | 2.586 |
| Wisc | 0.027 | 0.058 | 1.191 | 0.043 | 3.994 | 135.394 | 8.256 | 0.270 |

# E. Q2: Family-Specific Failure Modes

*Table 16.* Clustering performance summary by regime

| Regime | Rank | Method | Family | Median | Mean | #Datasets |
|---|---|---|---|---|---|---|
| A | 1 | Spectral | Spectral | 0.11523569 | 0.09938005 | 4 |
| A | 2 | K-means | classical_center | 0.10256874 | 0.09222711 | 4 |
| A | 3 | GMM | classical_center | 0.09693738 | 0.09035207 | 4 |
| A | 4 | AggWard | linkage | 0.09571605 | 0.08589508 | 4 |
| A | 5 | IDEC | deep-k-center | 0.08339325 | 0.07849580 | 4 |
| A | 6 | DCN | deep-k-center | 0.05114919 | 0.05827962 | 4 |
| A | 7 | SHADE | deep-density | 0.03619983 | 0.03308379 | 4 |
| A | 8 | HDBSCAN | density | 0.02318317 | 0.01996973 | 4 |
| A | 9 | DDC | deep-pipeline | 0.01931248 | 0.03708609 | 4 |
| A | 10 | AggAverage | linkage | 0.00388507 | 0.01106011 | 4 |
| B | 1 | DDC | deep-pipeline | 0.69508802 | 0.67258995 | 3 |
| B | 2 | Spectral | Spectral | 0.63083666 | 0.65343677 | 3 |
| B | 3 | DCN | deep-k-center | 0.57985437 | 0.61195484 | 3 |
| B | 4 | IDEC | deep-k-center | 0.57779703 | 0.64600005 | 3 |
| B | 5 | AggWard | linkage | 0.56782924 | 0.59638175 | 3 |
| B | 6 | K-means | classical_center | 0.49065938 | 0.48997221 | 3 |
| B | 7 | GMM | classical_center | 0.38395884 | 0.43772981 | 3 |
| B | 8 | SHADE | deep-density | 0.34939495 | 0.33417424 | 3 |
| B | 9 | HDBSCAN | density | 0.05239421 | 0.12454511 | 3 |
| B | 10 | AggAverage | linkage | 0.00079756 | 0.08016998 | 3 |
| C | 1 | Spectral | Spectral | 0.78236759 | 0.76567492 | 5 |
| C | 2 | AggWard | linkage | 0.72813631 | 0.73224767 | 5 |
| C | 3 | DDC | deep-pipeline | 0.72720126 | 0.64013834 | 5 |
| C | 4 | DCN | deep-k-center | 0.70847964 | 0.68077196 | 5 |
| C | 5 | IDEC | deep-k-center | 0.70636725 | 0.64467539 | 5 |
| C | 6 | GMM | classical_center | 0.70185817 | 0.71442991 | 5 |
| C | 7 | K-means | classical_center | 0.68198381 | 0.71224437 | 5 |
| C | 8 | AggAverage | linkage | 0.66727110 | 0.65774866 | 5 |
| C | 9 | SHADE | deep-density | 0.45361079 | 0.45819554 | 5 |
| C | 10 | HDBSCAN | density | 0.45139736 | 0.43475063 | 5 |

*Table 17.* Best-performing method per family and regime

| Regime | Family | Best Method | Median | Mean | #Datasets |
|---|---|---|---|---|---|
| A | classical_center | K-means | 0.10256874 | 0.09222711 | 4 |
| A | deep-density | SHADE | 0.03619983 | 0.03308379 | 4 |
| A | deep-k-center | IDEC | 0.08339325 | 0.07849580 | 4 |
| A | deep-pipeline | DDC | 0.01931248 | 0.03708609 | 4 |
| A | density | HDBSCAN | 0.02318317 | 0.01996973 | 4 |
| A | linkage | AggWard | 0.09571605 | 0.08589508 | 4 |
| A | Spectral | Spectral | 0.11523569 | 0.09938005 | 4 |
| B | classical_center | K-means | 0.49065938 | 0.48997221 | 3 |
| B | deep-density | SHADE | 0.34939495 | 0.33417424 | 3 |
| B | deep-k-center | DCN | 0.57985437 | 0.61195484 | 3 |
| B | deep-pipeline | DDC | 0.69508802 | 0.67258995 | 3 |
| B | density | HDBSCAN | 0.05239421 | 0.12454511 | 3 |
| B | linkage | AggWard | 0.56782924 | 0.59638175 | 3 |
| B | Spectral | Spectral | 0.63083666 | 0.65343677 | 3 |
| C | classical_center | GMM | 0.70185817 | 0.71442991 | 5 |
| C | deep-density | SHADE | 0.45361079 | 0.45819554 | 5 |
| C | deep-k-center | DCN | 0.70847964 | 0.68077196 | 5 |
| C | deep-pipeline | DDC | 0.72720126 | 0.64013834 | 5 |
| C | density | HDBSCAN | 0.45139736 | 0.43475063 | 5 |
| C | linkage | AggWard | 0.72813631 | 0.73224767 | 5 |
| C | Spectral | Spectral | 0.78236759 | 0.76567492 | 5 |

*Table 18.* Dataset-level family median clustering performance. These values are reported for descriptive and diagnostic purposes and are not used as inputs to the regime-level method or family rankings in Q2

| Dataset | Reg. | Classical | Linkage | Density | Spectral | Deep-$k$ | Deep-Dens. | Deep-Pipe. |
|---|---|---|---|---|---|---|---|---|
| CIFAR-10 | A | 0.0733 | 0.0357 | 0.0165 | 0.0835 | 0.0592 | 0.0553 | 0.0000 |
| CIFAR-100 | A | 0.1608 | 0.0927 | 0.0080 | 0.1612 | 0.1368 | 0.0457 | 0.0346 |
| STL-10 | A | 0.1262 | 0.0639 | 0.0152 | 0.1469 | 0.0753 | 0.0267 | 0.1097 |
| SVHN | A | 0.0049 | 0.0016 | 0.0003 | 0.0059 | 0.0022 | 0.0046 | 0.0041 |
| FMNIST | B | 0.5401 | 0.2673 | 0.0262 | 0.6308 | 0.5788 | 0.3494 | 0.4288 |
| KMNIST | B | 0.4142 | 0.2843 | 0.0077 | 0.5770 | 0.5571 | 0.2324 | 0.6951 |
| MNIST | B | 0.4373 | 0.4632 | 0.1530 | 0.7525 | 0.7510 | 0.4207 | 0.8938 |
| Dermatology | C | 0.8537 | 0.8454 | 0.2257 | 0.8824 | 0.7301 | 0.4536 | 0.7467 |
| Ecoli | C | 0.6141 | 0.6353 | 0.1183 | 0.5757 | 0.5457 | 0.3732 | 0.5332 |
| Pendigits | C | 0.6919 | 0.6899 | 0.3897 | 0.7824 | 0.7074 | 0.7490 | 0.8381 |
| Texture | C | 0.7691 | 0.5680 | 0.3341 | 0.7677 | 0.5795 | 0.5159 | 0.7272 |
| Wisc | C | 0.6378 | 0.7365 | 0.4001 | 0.8203 | 0.7508 | 0.1994 | 0.3555 |

*Table 19.* Clustering results on default setting.

| Dataset | AggAverage | AggWard | DBSCAN | DCN | DDC | GMM | HDBSCAN | IDEC | K-means | SHADE | Spectral |
|---|---|---|---|---|---|---|---|---|---|---|---|
| CIFAR-10 | 0.00052 | 0.07081 | 0.00000 | 0.04964 | 0.00000 | 0.06767 | 0.03294 | 0.06883 | 0.07893 | 0.05534 | 0.08353 |
| CIFAR-100 | 0.03595 | 0.14952 | 0.00000 | 0.13011 | 0.03455 | 0.16045 | 0.01594 | 0.14356 | 0.16110 | 0.04570 | 0.16116 |
| FMNIST | 0.00057 | 0.53401 | 0.00000 | 0.57985 | 0.42884 | 0.56856 | 0.05239 | 0.57772 | 0.51158 | 0.34939 | 0.63084 |
| KMNIST | 0.00080 | 0.56783 | 0.00000 | 0.53642 | 0.69509 | 0.36067 | 0.01533 | 0.57780 | 0.46768 | 0.23240 | 0.57702 |
| MNIST | 0.23914 | 0.68731 | 0.00000 | 0.71959 | 0.89385 | 0.38396 | 0.30591 | 0.78249 | 0.49066 | 0.42073 | 0.75246 |
| STL-10 | 0.00724 | 0.12062 | 0.00000 | 0.05266 | 0.10972 | 0.12621 | 0.03043 | 0.09796 | 0.12621 | 0.02670 | 0.14694 |
| SVHN | 0.00053 | 0.00262 | 0.00000 | 0.00071 | 0.00407 | 0.00709 | 0.00057 | 0.00364 | 0.00267 | 0.00460 | 0.00589 |
| Dermatology | 0.82685 | 0.86390 | 0.00000 | 0.74291 | 0.74675 | 0.82439 | 0.45140 | 0.71736 | 0.88307 | 0.45361 | 0.88239 |
| Ecoli | 0.66727 | 0.60325 | 0.11832 | 0.55341 | 0.53316 | 0.60841 | 0.11832 | 0.53798 | 0.61987 | 0.37316 | 0.57565 |
| Pendigits | 0.65168 | 0.72814 | 0.03804 | 0.70848 | 0.83805 | 0.70186 | 0.74130 | 0.70637 | 0.68198 | 0.74897 | 0.78237 |
| Texture | 0.46979 | 0.66612 | 0.00220 | 0.66854 | 0.72720 | 0.90468 | 0.66598 | 0.49051 | 0.63357 | 0.51588 | 0.76767 |
| Wisc | 0.67314 | 0.79983 | 0.60353 | 0.73052 | 0.35553 | 0.53281 | 0.19676 | 0.77116 | 0.74272 | 0.19935 | 0.82029 |

**E.1. Sanity checks on synthetic data**

To verify that CHB's regime assignments align with known clustering pathologies, we apply the fingerprint with frozen thresholds to three synthetic datasets whose structural properties are controlled by construction.

**Well-separated blobs (positive control).** Five isotropic Gaussian clusters in $\mathbb{R}^{10}$ ($n=2000$, $\sigma=0.5$, centers spread over $[-15, 15]^{10}$). All separation indicators are benign ($\mathcal{S}_1=0$, $\mathcal{S}_2=0$, $\mathcal{S}_3=21.2$), topology evidence is low ($\mathcal{T}_{\text{evid}}=6.1$), and CHB assigns **Regime C**. $k$-means recovers the partition perfectly (NMI=1.0), confirming that Regime C correctly identifies an easy, broadly recoverable instance.

**Anisotropic clusters (cohesion mismatch).** Four well-separated but strongly elongated elliptical clusters in $\mathbb{R}^2$ ($n=2000$). The separability gate passes easily ($\mathcal{S}_1=0.05$, $\mathcal{S}_2=0.04$, $\mathcal{S}_3=18.3$) and topology evidence remains low ($\mathcal{T}_{\text{evid}}=3.1$), so CHB assigns **Regime C**. However, linear elongation is extreme ($\mathcal{C}_2=87.1$), and $k$-means achieves only NMI=0.59. CHB identifies the bottleneck as cohesion mismatch.

**Overlapping blobs (negative control).** Five Gaussian clusters in $\mathbb{R}^{10}$ with high variance ($\sigma=4.0$) and tightly spaced centers ($[-5, 5]^{10}$), producing heavy inter-cluster overlap. The separability gate fails, and CHB assigns **Regime A**. $k$-means achieves low NMI, consistent with the near-universal failure predicted by separability collapse.

Together, these three cases confirm that CHB's regime assignments track the intended failure-mode semantics: Regime C for recoverable instances (with $\mathcal{C}_2$ identifying within-regime mismatch), and Regime A for genuine separability collapse.

*Table 20.* Best variant score per dataset and clustering family. For each dataset, we report the maximum score achieved by each family across all representation variants. The best family per dataset is highlighted in bold. *This is a per-family oracle over representations*

| Dataset | Center | Deep | Deep-Density | Deep-Pipeline | Density | Linkage | Spectral |
|---|---|---|---|---|---|---|---|
| CIFAR-10 | 0.102 | 0.097 | 0.126 | 0.034 | **0.176** | 0.103 | 0.103 |
| CIFAR-100 | 0.182 | 0.181 | 0.244 | 0.031 | **0.351** | 0.182 | 0.180 |
| FMNIST | 0.684 | 0.669 | 0.620 | 0.527 | 0.620 | 0.680 | 0.647 |
| KMNIST | 0.689 | 0.659 | **0.697** | 0.678 | 0.638 | 0.634 | 0.603 |
| MNIST | **0.919** | 0.901 | 0.911 | 0.840 | 0.779 | 0.905 | 0.786 |
| STL-10 | 0.164 | 0.163 | 0.162 | 0.171 | **0.188** | 0.165 | 0.169 |
| SVHN | 0.019 | 0.016 | 0.054 | 0.007 | **0.114** | 0.018 | 0.015 |

*Table 21.* Side-by-side comparison of clustering performance (NMI ↑) using **global default best** (Default-Best; base_score with agg=best) versus the **best learned representation** (Best-Rep; max variant_score over representations, agg=best) for datasets initially in Regime A or B (base/raw), before representation transitions.

| Dataset | Default-Best (agg=best; base_score) | | | | | | | Best-Rep (agg=best; max variant_score) | | | | | | |
|---|---|---|---|---|---|---|---|---|---|---|---|---|---|---|
| | Center | Linkage | Density | Spectral | Deep-$k$ | Deep-Dens | Deep-Pipe | Center | Linkage | Density | Spectral | Deep-$k$ | Deep-Dens | Deep-Pipe |
| **Regime A** | | | | | | | | | | | | | | |
| CIFAR-10 | 0.079 | 0.071 | 0.033 | 0.084 | 0.069 | 0.055 | 0.000 | 0.102 | 0.103 | **0.176** | 0.103 | 0.097 | 0.126 | 0.034 |
| CIFAR-100 | 0.161 | 0.150 | 0.016 | 0.161 | 0.144 | 0.046 | 0.035 | 0.182 | 0.182 | **0.351** | 0.180 | 0.181 | 0.244 | 0.031 |
| STL-10 | 0.126 | 0.121 | 0.030 | 0.147 | 0.098 | 0.027 | 0.110 | 0.164 | 0.165 | **0.188** | 0.169 | 0.163 | 0.162 | 0.171 |
| SVHN | 0.007 | 0.003 | 0.001 | 0.006 | 0.004 | 0.005 | 0.004 | 0.019 | 0.018 | **0.114** | 0.015 | 0.016 | 0.054 | 0.007 |
| **Regime B** | | | | | | | | | | | | | | |
| FMNIST | 0.569 | 0.534 | 0.052 | 0.631 | 0.580 | 0.349 | 0.429 | 0.684 | 0.680 | **0.620** | 0.647 | 0.669 | 0.620 | 0.527 |
| KMNIST | 0.468 | 0.568 | 0.015 | 0.577 | 0.578 | 0.232 | 0.695 | 0.689 | 0.634 | **0.638** | 0.603 | 0.659 | 0.697 | 0.678 |
| MNIST | 0.491 | 0.687 | 0.306 | 0.752 | 0.782 | 0.421 | 0.894 | **0.919** | 0.905 | 0.779 | 0.795 | 0.901 | 0.911 | 0.840 |

# F. Q3 - Representation shift

**Movement in CHB space under representation learning.** Tables 22-28 report the movement of Image data (Table 8) in CHB space induced by different representation variants. We evaluate multiple alternative representations of the same dataset, including UMAP embeddings with target dimensionalities $\{4, 64, 128, 784\}$, and the latent representation produced by DDC. Each variant is treated as a separate dataset–representation instance and analyzed using identical default CHB settings. For each representation, we compute the CHB hardness fingerprint and report raw descriptor differences $\Delta = \mathbf{h}(\mathcal{D}_{\text{variant}}) - \mathbf{h}(\mathcal{D}_{\text{base}})$ relative to the base (raw) representation. All CHB coordinates are computed at the cluster level and then aggregated to the dataset level using a cluster-size–weighted 10% trimmed mean. The reported deltas therefore, quantify representation-induced structural shifts along the separability ($\mathcal{S}$), cohesion ($\mathcal{C}$), and topology ($\mathcal{T}$) axes, independent of clustering algorithm choice or descriptor hyperparameter tuning. The *Trans.* column indicates the corresponding regime transition under the fixed CHB regime assignment rule.

*Table 22.* CIFAR-10: Movement in CHB space induced by representation learning. We report raw axis deltas (variant minus base) using the $\mathcal{S}/\mathcal{C}/\mathcal{T}$ notation.

| Variant | Trans. | $\Delta\mathcal{S}_1$ | $\Delta\mathcal{S}_2$ | $\Delta\mathcal{S}_3$ | $\Delta\mathcal{C}_2$ | $\Delta\mathcal{C}_1$ | $\Delta\mathcal{T}_1$ | $\Delta\mathcal{T}_2$ | $\Delta\mathcal{T}_3$ |
|---|---|---|---|---|---|---|---|---|---|
| DDC | A→A | 0.020 | 0.365 | -0.501 | -1048.6 | -0.041 | -32787.6 | -108.9 | -7.108 |
| UMAP 128 | A→A | 0.044 | 0.382 | -0.298 | -917.6 | -0.066 | -31136.0 | -23.4 | 3.503 |
| UMAP 4 | A→A | 0.062 | 0.402 | -0.338 | -1046.9 | -0.071 | -32630.6 | -99.9 | -6.059 |
| UMAP 64 | A→A | 0.047 | 0.385 | -0.302 | -978.9 | -0.051 | -31683.8 | -48.6 | -0.188 |
| UMAP 784 | A→A | 0.050 | 0.389 | -0.308 | -305.5 | -0.085 | -29610.5 | 45.4 | 11.853 |

*Table 23.* CIFAR-100: Movement in CHB space induced by representation learning. We report raw axis deltas (variant minus base) using the $\mathcal{S}/\mathcal{C}/\mathcal{T}$ notation.

| Variant | Trans. | $\Delta\mathcal{S}_1$ | $\Delta\mathcal{S}_2$ | $\Delta\mathcal{S}_3$ | $\Delta\mathcal{C}_2$ | $\Delta\mathcal{C}_1$ | $\Delta\mathcal{T}_1$ | $\Delta\mathcal{T}_2$ | $\Delta\mathcal{T}_3$ |
|---|---|---|---|---|---|---|---|---|---|
| DDC | A→A | 0.019 | 0.439 | -0.642 | -211.6 | -0.066 | -19843.2 | -151.2 | -14.042 |
| UMAP 128 | A→A | 0.020 | 0.433 | -0.476 | -51.9 | -0.125 | -18795.1 | -77.1 | -6.143 |
| UMAP 4 | A→A | 0.026 | 0.443 | -0.519 | -208.3 | -0.137 | -19729.6 | -142.2 | -13.059 |
| UMAP 64 | A→A | 0.021 | 0.435 | -0.479 | -120.2 | -0.123 | -19146.8 | -101.8 | -9.031 |
| UMAP 784 | A→A | 0.019 | 0.433 | -0.476 | 496.5 | -0.124 | -17673.8 | 2.0 | 2.312 |

*Table 24.* STL-10: Movement in CHB space induced by representation learning. We report raw axis deltas (variant minus base) using the $\mathcal{S}/\mathcal{C}/\mathcal{T}$ notation.

| Variant | Trans. | $\Delta\mathcal{S}_1$ | $\Delta\mathcal{S}_2$ | $\Delta\mathcal{S}_3$ | $\Delta\mathcal{C}_2$ | $\Delta\mathcal{C}_1$ | $\Delta\mathcal{T}_1$ | $\Delta\mathcal{T}_2$ | $\Delta\mathcal{T}_3$ |
|---|---|---|---|---|---|---|---|---|---|
| DDC | A→A | -0.019 | 0.394 | -0.549 | -102.2 | -0.037 | -68651.0 | -218.7 | -10.840 |
| UMAP 128 | A→A | 0.009 | 0.415 | -0.437 | 56.5 | -0.053 | -67903.8 | -165.8 | -7.060 |
| UMAP 4 | A→A | 0.012 | 0.419 | -0.458 | -98.4 | -0.096 | -68570.9 | -213.1 | -10.239 |
| UMAP 64 | A→A | 0.010 | 0.415 | -0.437 | -21.1 | -0.008 | -68142.9 | -184.0 | -8.795 |
| UMAP 784 | A→A | 0.006 | 0.411 | -0.433 | 524.1 | -0.022 | -66724.8 | -85.0 | -1.307 |

*Table 25.* SVHN: Movement in CHB space induced by representation learning. We report raw axis deltas (variant minus base) using the $\mathcal{S}/\mathcal{C}/\mathcal{T}$ notation.

| Variant | Trans. | $\Delta\mathcal{S}_1$ | $\Delta\mathcal{S}_2$ | $\Delta\mathcal{S}_3$ | $\Delta\mathcal{C}_2$ | $\Delta\mathcal{C}_1$ | $\Delta\mathcal{T}_1$ | $\Delta\mathcal{T}_2$ | $\Delta\mathcal{T}_3$ |
|---|---|---|---|---|---|---|---|---|---|
| DDC | A→A | 0.173 | 0.388 | -0.574 | -4206.9 | -0.133 | -23811.7 | -177.9 | -11.701 |
| UMAP 128 | A→A | 0.118 | 0.325 | -0.310 | -4145.0 | -0.150 | -22247.5 | -84.3 | -2.785 |
| UMAP 4 | A→A | 0.162 | 0.373 | -0.342 | -4205.0 | -0.161 | -23661.2 | -168.2 | -10.715 |
| UMAP 64 | A→A | 0.127 | 0.334 | -0.310 | -4176.0 | -0.159 | -22734.5 | -113.2 | -5.509 |
| UMAP 784 | A→A | 0.123 | 0.329 | -0.315 | -3325.7 | -0.172 | -21993.4 | -70.7 | -3.205 |

*Table 26.* FMNIST: Movement in CHB space induced by representation learning. We report raw axis deltas (variant minus base) using the $\mathcal{S}/\mathcal{C}/\mathcal{T}$ notation.

| Variant | Trans. | $\Delta\mathcal{S}_1$ | $\Delta\mathcal{S}_2$ | $\Delta\mathcal{S}_3$ | $\Delta\mathcal{C}_2$ | $\Delta\mathcal{C}_1$ | $\Delta\mathcal{T}_1$ | $\Delta\mathcal{T}_2$ | $\Delta\mathcal{T}_3$ |
|---|---|---|---|---|---|---|---|---|---|
| DDC | B→C | -0.010 | 0.025 | 1.518 | -235.0 | 0.024 | -11804.4 | -135.2 | -16.803 |
| UMAP 128 | B→C | 0.029 | 0.064 | 0.245 | -106.0 | -0.018 | -11006.3 | -99.9 | -15.188 |
| UMAP 4 | B→C | 0.037 | 0.072 | 0.400 | -221.5 | 0.024 | -11778.4 | -134.3 | -16.741 |
| UMAP 64 | B→C | 0.031 | 0.066 | 0.257 | -167.6 | -0.019 | -11277.1 | -111.2 | -15.602 |
| UMAP 784 | B→C | 0.028 | 0.064 | 0.236 | 487.9 | -0.009 | -10119.9 | -60.0 | -13.845 |

*Table 27.* KMNIST: Movement in CHB space induced by representation learning. We report raw axis deltas (variant minus base) using the $\mathcal{S}/\mathcal{C}/\mathcal{T}$ notation.

| Variant | Trans. | $\Delta\mathcal{S}_1$ | $\Delta\mathcal{S}_2$ | $\Delta\mathcal{S}_3$ | $\Delta\mathcal{C}_2$ | $\Delta\mathcal{C}_1$ | $\Delta\mathcal{T}_1$ | $\Delta\mathcal{T}_2$ | $\Delta\mathcal{T}_3$ |
|---|---|---|---|---|---|---|---|---|---|
| DDC | B→C | -0.024 | -0.026 | 4.476 | -100.9 | -0.028 | -17194.4 | -179.9 | -22.851 |
| UMAP 128 | B→C | -0.035 | -0.040 | 1.683 | 60.9 | -0.083 | -16509.5 | -154.6 | -21.388 |
| UMAP 4 | B→C | -0.035 | -0.038 | 2.721 | -92.2 | -0.094 | -17172.1 | -179.5 | -22.832 |
| UMAP 64 | B→C | -0.035 | -0.043 | 1.764 | -10.4 | -0.116 | -16749.3 | -163.4 | -22.005 |
| UMAP 784 | B→C | -0.034 | -0.042 | 1.706 | 805.8 | -0.109 | -15421.1 | -112.9 | -19.355 |

*Table 28.* MNIST: Movement in CHB space induced by representation learning. We report raw axis deltas (variant minus base) using the $\mathcal{S}/\mathcal{C}/\mathcal{T}$ notation.

| Variant | Trans. | $\Delta\mathcal{S}_1$ | $\Delta\mathcal{S}_2$ | $\Delta\mathcal{S}_3$ | $\Delta\mathcal{C}_2$ | $\Delta\mathcal{C}_1$ | $\Delta\mathcal{T}_1$ | $\Delta\mathcal{T}_2$ | $\Delta\mathcal{T}_3$ |
|---|---|---|---|---|---|---|---|---|---|
| DDC | B→C | -0.047 | -0.065 | 5.386 | -65.3 | 0.199 | -12082.8 | -178.4 | -24.541 |
| UMAP 2D | B→C | -0.017 | -0.035 | 3.044 | -64.6 | 0.187 | -12088.0 | -179.0 | -24.541 |
| UMAP 128 | B→C | -0.037 | -0.055 | 1.653 | 79.6 | 0.178 | -11519.4 | -153.3 | -22.708 |
| UMAP 4 | B→C | -0.037 | -0.055 | 1.956 | -63.0 | 0.148 | -12056.6 | -177.6 | -24.462 |
| UMAP 64 | B→C | -0.037 | -0.055 | 1.662 | 18.9 | 0.204 | -11687.2 | -160.9 | -23.262 |
| UMAP 784 | B→C | -0.037 | -0.055 | 1.678 | 1059.1 | 0.168 | -10912.0 | -129.8 | -21.752 |

*Table 29.* CHB regime trajectories under a CLIP → UMAP pipeline. *"Sep. rescued?" indicates whether the separability gate passes after the CLIP stage. "Trajectory" tracks regimes from raw features to the final embedding. ΔNMI is the net change in k-means NMI over the full pipeline.*

| Dataset | Sep. rescued? | Trajectory | ΔNMI |
|---|---|---|---|
| CIFAR-10 | ✓ | A→B→C | +.713 |
| MNIST | (already ok) | B→C | +.347 |
| SVHN | × | A→A | −.044 |
| AG News | (strengthened) | B→C | +.192 |
| 20 NG | × | A→A | +.046 |

## F.1. Stress-Testing CHB on a Modern Representation Pipeline

As part of an expanded stress test, we evaluate CHB on a modern foundation-model pipeline (CLIP (Radford et al., 2021) → UMAP) and ask whether regime assignments remain informative when applied after each stage. If CHB is a useful diagnostic, it should identify structural bottlenecks at intermediate pipeline stages—not only at the endpoint (Table 29).

We apply OpenCLIP ViT-B/32 followed by UMAP to image and text datasets, computing CHB fingerprints after each stage with frozen thresholds and no re-tuning. For each dataset we track the regime trajectory from raw features through the final embedding and report the net change in $k$-means NMI.

**Success case: CIFAR-10 (A→B→C).** Raw CIFAR-10 sits in Regime A with $k$-means NMI $= .08$. After CLIP, the separability gate passes and CHB assigns Regime B. After UMAP, CHB assigns Regime C and $k$-means scores .80. CHB diagnoses the progression at each stage: the structural bottleneck shifts from failed separability to residual topological complexity to near-complete resolution.

**Failure case: SVHN (A→A).** CIFAR-10 and SVHN both start in Regime A with nearly identical raw overlap, yet the same pipeline yields $+.713$ and $−.044$ NMI respectively. CHB explains the divergence at the intermediate stage: after CLIP, CIFAR-10 passes the separability gate while SVHN does not. The final NMI difference is already anticipated by the regime assignment after CLIP—before any downstream clustering is run.

**Cross-modality consistency.** AG News (text, $K=4$) (Zhang et al., 2015) starts in Regime B; after the full pipeline it reaches Regime C with $+.192$ NMI. 20 Newsgroups (text, $K=20$) starts in Regime A and remains there, with only a marginal net gain. The regime logic transfers across modalities without modification: datasets whose separability gate remains closed after CLIP show limited downstream improvement regardless of further processing.

**Diagnostic value.** Across all five cases, the regime assignment after CLIP correctly separates datasets that will benefit from the remainder of the pipeline from those that will not. While NMI reports only the endpoint, CHB provides an earlier and more informative signal by identifying the structural bottleneck at each stage. This confirms that CHB remains a useful diagnostic for modern representation pipelines, not only for raw-feature benchmarks.

## F.2. Tabular generalization

To test whether regime boundaries remain stable beyond image data, we apply CHB with frozen thresholds to two additional tabular datasets. Eucalyptus fails the separability gate on all three indicators and $k$-means achieves NMI $= .08$, consistent with Regime A. Cardiotocography passes the gate with strong margins and low topology evidence, landing in Regime C with $k$-means NMI $= .91$. Together with the text results above, the regime structure generalizes across three modalities without re-tuning.

# G. Ablation Study: Separation, Cohesion, Topology

## G.1. Why CHB needs all three blocks (Separation, Cohesion, Topology): regime boundaries and Q1–Q3 triangulation

CHB's fingerprint (Separation, Cohesion, Topology) is a minimal mechanism decomposition aligned with the separability-first hierarchy (Q1), regime-conditioned rank reversals (Q2), and representation auditing as movement in descriptor space (Q3). Table 1 maps theoretical conditions to descriptor blocks: local identifiability $\leftrightarrow S$, structural alignment $\leftrightarrow T$, and scale compatibility $\leftrightarrow C$. The descriptor-only PCA in Q1 further supports this semantics: PC1 aligns with the separability gate, PC2 contrasts topology-mismatch against heterogeneity (driven by $\mathcal{T}_2 - \mathcal{T}_3$ vs. $\mathcal{C}_1, \mathcal{C}_2$), and PC3 captures within-regime severity (primarily $\mathcal{C}_1, \mathcal{C}_2$).

**Empirical block-drop PCA: each block preserves a distinct regime boundary.**    To stress-test non-redundancy, we repeat the descriptor-only PCA after removing one block. Two collapses occur: (i) removing $T$ (PCA on $(\mathcal{S}_1 - \mathcal{S}_3, C)$) merges digit instances with tabular instances Fig. 7, collapsing the post-gate distinction between topology-mismatch (Regime B) and scale heterogeneity (Regime C); (ii) removing $\mathcal{S}_1 - \mathcal{S}_3$ (PCA on $(C, T)$) merges digit instances with natural-image pixels, collapsing the gate distinction between separability collapse (Regime A) and usable-separation mismatch (Regime B) Fig. 6. Thus, each block corresponds to a different regime boundary: $S$ separates A from non-A, while $T$ separates B from C given a passing gate.

**Why $\mathcal{S}_1 - \mathcal{S}_3$ is necessary (the gate; regime A and limits of algorithm choice).**    Without $S$, topology indicators are not self-interpreting: under separability collapse, corrupted neighborhoods inflate connectivity-scale summaries and can destabilize higher-order PH, producing topology-like signals that are largely artifacts of the broken gate. This is why Q2 reports near-universal failure in Regime A and why Q3 shows that mismatch reduction without separability rescue yields only bounded, often family-conditional gains on natural-image pixels.

**Why topology is necessary (it prevents mixing mismatch-driven digits with heterogeneity-driven tabular).**    Digits (Regime B) and tabular datasets (Regime C) both pass the gate (low $\mathcal{S}_1, \mathcal{S}_2$, high $\mathcal{S}_3$), and can overlap in cohesion. The distinguishing mechanism is persistent nontrivial topology (large $\mathcal{T}_1 - \mathcal{T}_3$), which defines mismatch-driven difficulty. Removing $T$ therefore erases the mismatch–vs–heterogeneity axis (PC2), making B and C indistinguishable.

This boundary is operationally critical: Q2 exhibits a regime-dependent family reversal precisely across B vs C (deep-pipeline dominates in B, while Spectral dominates in C). Without $T$, this reversal is no longer explainable from structure.

**Why $\mathcal{C}_1, \mathcal{C}_2$ is necessary (heterogeneity and within-regime severity accounting).**    $C$ captures non-exchangeable cluster resolutions: even when separability is strong, and topology is simple, multi-scale density and anisotropy induce merge–split instability for single-resolution procedures. This is the defining mechanism of Regime C and also the primary carrier of within-regime severity (PC3), explaining why separability can remain stable while hardness changes materially (e.g., MNIST→KMNIST in Q1).

**Q3 link: topology is the axis representation learning simplifies when classical clustering starts to work.**    Q3 identifies mismatch correction as large decreases in $T_{1:3}$ under usable separation, and empirically large gains concentrate in B→C transitions for digit datasets. In these cases, representations reduce topological complexity and often improve local separability, after which classical center/linkage objectives become well-specified and improve broadly. In contrast, for A→A negative controls, $T$ can decrease without consistent separability rescue, yielding only family-conditional improvements.

**Takeaway.**    The full $(S, C, T)$ fingerprint is non-redundant: it is the minimal decomposition that preserves (i) gate detection (A vs non-A), (ii) post-gate mode identification (B vs C), and (iii) within-regime severity accounting, enabling Q2 mechanism-conditioned rank reversals and Q3 structural attribution.

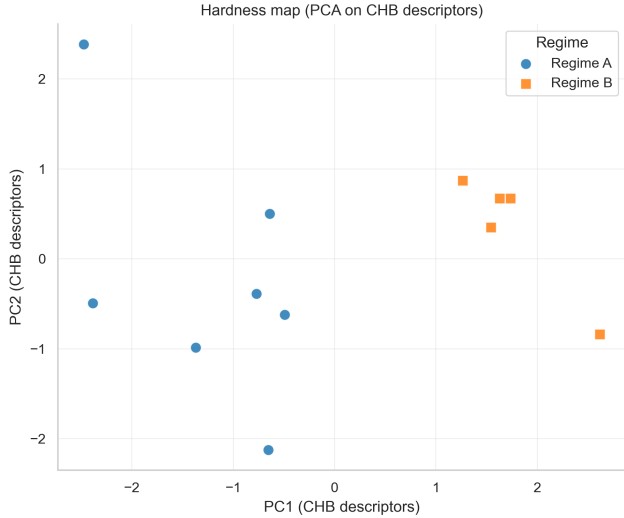

*Figure 6.* **Hardness map without Separation** $\mathcal{S}_1 - \mathcal{S}_3$ **(descriptor-only PCA).** Each point is a base/raw dataset embedded by PCA of standardized CHB descriptors Colors denote regimes A/B/C.

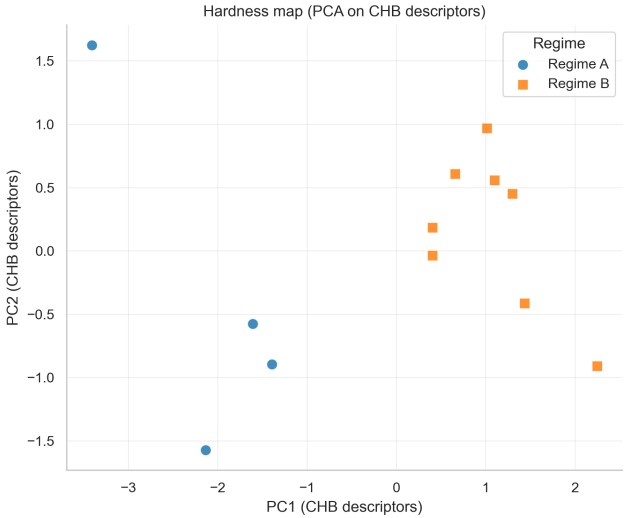

*Figure 7.* **Hardness map without Topology** $\mathcal{T}_1 - \mathcal{T}_3$ **(descriptor-only PCA).** Each point is a base/raw dataset embedded by PCA of standardized CHB descriptors Colors denote regimes A/B/C.

*Table 30.* Best NMI per dataset and method. "–" indicates that an error occurred during execution.

| Dataset | K-means | AggWard | GMM | Spectral | DCN | IDEC | DDC | HDBSCAN |
|---|---|---|---|---|---|---|---|---|
| MNIST | 0.491 | 0.687 | – | 0.752 | 0.720 | 0.799 | 0.898 | 0.306 |
| FMNIST | 0.512 | 0.534 | – | 0.631 | 0.585 | 0.631 | 0.664 | 0.089 |
| KMNIST | 0.468 | 0.568 | – | 0.580 | 0.536 | 0.583 | 0.695 | 0.044 |
| CIFAR-10 | 0.079 | 0.071 | – | 0.090 | 0.076 | 0.079 | 0.020 | 0.057 |
| CIFAR-100 | 0.161 | 0.150 | – | 0.171 | 0.164 | 0.145 | 0.035 | 0.021 |
| SVHN | 0.003 | 0.003 | – | 0.007 | 0.001 | 0.007 | 0.004 | 0.000 |
| STL-10 | 0.126 | 0.121 | – | 0.147 | 0.068 | 0.125 | 0.110 | 0.030 |
| Pendigits | 0.682 | 0.728 | 0.721 | 0.804 | 0.732 | 0.733 | 0.839 | 0.741 |
| optdigits | 0.758 | 0.825 | 0.761 | 0.896 | 0.814 | 0.843 | 0.923 | 0.524 |
| Wisc | 0.743 | 0.800 | 0.721 | 0.820 | 0.731 | 0.771 | 0.356 | 0.210 |
| Dermatology | 0.883 | 0.864 | 0.910 | 0.883 | 0.915 | 0.914 | 0.904 | 0.600 |
| Ecoli | 0.620 | 0.603 | 0.608 | 0.577 | 0.594 | 0.587 | 0.681 | 0.314 |
| isolet | 0.737 | 0.758 | 0.740 | 0.769 | 0.772 | 0.674 | – | 0.131 |
| Texture | 0.634 | 0.666 | 0.905 | 0.768 | 0.700 | 0.502 | 0.727 | 0.661 |
| letter | 0.359 | 0.393 | 0.488 | 0.428 | 0.443 | 0.312 | 0.443 | 0.479 |
| robot-failure-lp1 | 0.169 | 0.211 | 0.234 | 0.418 | 0.412 | 0.383 | 0.463 | 0.000 |

*Table 31.* Side-by-side comparison of best NMI (↑) obtained from raw input with hyperparameter optimization (Raw) and from the best learned representation (Repr). For each dataset and clustering family, we report the maximum NMI achieved across all variants.

| | Raw Input | | | | | | | Best Representation | | | | | | |
|---|---|---|---|---|---|---|---|---|---|---|---|---|---|---|
| Dataset | Center | Deep | Deep-Dens | Deep-Pipe | Density | Linkage | Spectral | Center | Deep | Deep-Dens | Deep-Pipe | Density | Linkage | Spectral |
| MNIST | 0.491 | 0.759 | 0.601 | 0.898 | 0.306 | 0.687 | 0.752 | **0.919** | 0.901 | 0.911 | 0.840 | 0.779 | 0.905 | 0.786 |
| FMNIST | 0.512 | 0.631 | 0.415 | 0.664 | 0.089 | 0.534 | 0.631 | 0.684 | 0.669 | **0.620** | 0.527 | 0.620 | 0.680 | 0.647 |
| KMNIST | 0.468 | 0.583 | 0.497 | 0.695 | 0.044 | 0.568 | 0.580 | 0.689 | 0.659 | **0.697** | 0.678 | 0.638 | 0.634 | 0.603 |
| CIFAR-10 | 0.079 | 0.079 | 0.126 | 0.020 | 0.057 | 0.071 | 0.090 | 0.102 | 0.097 | 0.126 | 0.034 | **0.176** | 0.103 | 0.103 |
| CIFAR-100 | 0.161 | 0.145 | 0.244 | 0.035 | 0.021 | 0.150 | 0.171 | 0.182 | 0.181 | 0.244 | 0.031 | **0.351** | 0.182 | 0.180 |
| STL-10 | 0.126 | 0.125 | 0.162 | 0.110 | 0.030 | 0.121 | 0.147 | 0.164 | 0.163 | 0.162 | 0.171 | **0.188** | 0.165 | 0.169 |
| SVHN | 0.003 | 0.007 | 0.054 | 0.004 | 0.000 | 0.003 | 0.007 | 0.019 | 0.016 | 0.054 | 0.007 | **0.114** | 0.018 | 0.015 |

# H. Hyperparameter Tuning in CHB's Space

A reasonable concern is that CHB's regime conclusions (Q1–Q3) might reflect default settings or limited tuning rather than structural hardness. We therefore run an aggressive *oracle* stress test: for each dataset and method we select the best configuration from a large search and report best-of-search NMI (Table 30). This is not a deployable selection rule, but it probes whether tuning alone can erase the descriptor-defined hardness picture.

**Key findings.**

- **Regime structure survives oracle tuning.** Best-of-search performance improves within regimes but does not collapse the descriptor-defined stratification: Regime A remains low, Regime B rises substantially, and Regime C remains broadly recoverable (Table 30).

- **Family-level differences remain regime-dependent.** Extensive tuning preserves the qualitative regime-conditioned winner picture; tuning rarely turns a regime-limited dataset into a globally "easy" one. What changes most are *within-regime* gaps and occasional local reversals, rather than the overall regime ordering.

- **Representation sweeps reinforce the separability gate (Q3).** When representations do *not* repair local separability in Regime A, improvements remain bounded and often method-concentrated. When separability is usable (Regime B), representation changes can unlock broad, cross-family gains. Density-based methods show the sharpest representation sensitivity, exhibiting both dramatic gains and collapses depending on whether a stable scale/connectivity structure is induced.

**Takeaway.** Oracle hyperparameter search can yield meaningful *within-regime* gains, but it does not eliminate regime-conditioned hardness; escaping the hardest failures typically requires a representation change, consistent with CHB's Q1–Q3 mechanism view.

# I. Hyperparameter Stability Analysis

This section evaluates the robustness of all proposed dataset-level descriptors under controlled hyperparameter perturbations. Our goal is not to optimize performance, but to verify that the resulting difficulty signatures reflect intrinsic structural properties rather than artifacts of tuning.

All stability analyses are conducted at the dataset-summary level using the same cluster-size–weighted $10\%$ trimmed mean aggregation as in the main pipeline. For each descriptor, we report the median, interquartile range (IQR), and coefficient of variation (CV) across a small, deterministic hyperparameter grid.

## I.1. Stability Protocol

Hyperparameters are varied only along dimensions known to affect numerical estimation, while all semantic definitions are kept fixed. Grids are intentionally small and structured to reflect realistic practitioner choices rather than exhaustive search.

**Baseline.** All other baseline quantities (intrinsic dimension, Hopkins statistic, PCA-based features) are deterministic given the data.

**Cohesion and Topology.** For cohesion and topology descriptors, we isolate persistence-related sensitivity by varying only the lifetime threshold:

$$\tau \in \{0.005,\ 0.01,\ 0.02\}.$$

All geometric parameters (neighborhood fraction, MST approximation threshold, sampling budgets) are held fixed.

**Separation.** For separation, we jointly vary neighborhood scale, density resolution, and margin strictness using:

$$s_{12}^{\text{base}} \in \{10,\ 20,\ 40\}, \quad (k_{\text{density}}, k_{\text{graph}}) \in \{(10,8),(15,10),(20,12)\},$$

$$(q_{\text{margin}}, p_{\text{margin}}) \in \{(0.1,1),(0.25,3),(0.5,5)\}.$$

**Density.** Density robustness is evaluated by varying only the neighborhood size:

$$k_{\text{density}} \in \{10,\ 25,\ 50\}.$$

## I.2. Separation

Separation measures $\{\mathcal{S}_i\}$ assess inter-cluster interference and decision-boundary geometry.

$\mathcal{S}_1$ (overlap) and $\mathcal{S}_2$ (hubness infiltration) are highly stable across all datasets. Observed IQR values are near zero, with CV typically below $3\%$, indicating that local neighborhood scale has little effect on qualitative separation behavior.

$\mathcal{S}_3$ (margins) exhibits moderate sensitivity to margin parameters, as expected for distance-based criteria.

## I.3. Cohesion

Cohesion measures $\{\mathcal{C}_i\}$ quantify internal geometric regularity of clusters.

$\mathcal{C}_1$ (density-based summaries) vary smoothly with $k_{\text{density}}$, exhibiting moderate but bounded variability (typically CV below $10\%$). Importantly, internal multi-$k$ aggregation further reduces sensitivity to the exact neighborhood choice.

Linear elongation $\mathcal{C}_2$ is not density-driven. It depends exclusively on the covariance spectrum of each cluster and is therefore independent of density estimation, neighborhood size, or persistence thresholds. Empirically, $\mathcal{C}_2$ remains exactly invariant across all hyperparameter settings ($\text{IQR} = 0, \text{CV} = 0$).

## I.4. Topology

Topological descriptors capture higher-order structure via persistent homology.

Normalized PH0, PH1, and PH2 summaries remain stable across moderate lifetime thresholds. Resampled persistence measures ($\mathcal{T}_1, \mathcal{T}_2, \mathcal{T}_3$) show controlled variability, reflecting stochastic sampling rather than structural instability.

Crucially, while absolute persistence values vary with $\tau$, relative ordering across datasets is preserved, and median values remain consistent with default settings.

## I.5. Conclusion

Across all components, hyperparameter sensitivity is limited, interpretable, and localized. Geometric cohesion and separation descriptors are robust by construction, density acts as a controlled incohesion stressor, and topological variability reflects explicit lifetime filtering rather than instability. These results support the use of the proposed descriptors as reliable, mechanism-level characterizations rather than finely tuned scores.

*Table 32.* Comparison of key clustering and structural metrics across datasets with cluster internal metrics.

| Metric | CIFAR-10 | CIFAR-100 | FMNIST | KMNIST | MNIST | Pendigits | STL-10 | SVHN | Texture | Wisc | Dermatology | Ecoli |
|---|---|---|---|---|---|---|---|---|---|---|---|---|
| Silhouette (Rousseeuw, 1987) | -0.0558 | -0.1246 | 0.0261 | -0.0069 | -0.0363 | 0.1712 | -0.0417 | -0.0288 | 0.1958 | 0.5493 | 0.2073 | 0.2260 |
| Davies–Bouldin (Davies & Bouldin, 1979) | 12.07 | 8.75 | 3.71 | 5.01 | 4.60 | 2.25 | 13.07 | 34.15 | 2.04 | 0.86 | 1.64 | 1.59 |
| Calinski–Harabasz (Caliński & Harabasz, 1974) | 35.66 | 7.96 | 273.97 | 74.04 | 68.61 | 574.79 | 43.10 | 2.46 | 1442.19 | 776.62 | 85.67 | 67.52 |
| Intrinsic Dimensionality (Levina & Bickel, 2004) | 24.56 | 22.03 | 13.29 | 18.81 | 12.32 | 5.19 | 28.53 | 17.37 | 5.78 | 5.43 | 7.88 | 3.74 |
| Hopkins (Hopkins & Skellam, 1954) | 0.624 | 0.632 | 0.929 | 0.790 | 0.982 | 0.854 | 0.591 | 0.760 | 0.941 | 0.778 | 0.746 | 0.916 |

## J. Why internal Clusterability Metrics (CVI) Are Not a Separability or Regime Gate

A common but largely implicit practice in clustering benchmarks is to interpret standard internal validation metrics—such as Silhouette (Rousseeuw, 1987), Davies–Bouldin (DB) (Davies & Bouldin, 1979), and Calinski–Harabasz (CH) (Caliński & Harabasz, 1974)—as proxies for *separability*, and even as binary "gates" for whether meaningful cluster structure exists. Table 32 reports these baseline indices (together with intrinsic dimensionality and Hopkins) for our dataset suite. A typical heuristic is that near-zero or negative silhouette together with comparatively large DB indicates "separability collapse." We argue that such gating is unreliable at *both* boundaries required for hardness-aware evaluation. **Note.** Ground-truth labels are used only to evaluate classical cluster validity indices (CVIs) in order to assess the extent to which their values are biased or constrained by the true class structure. This analysis is purely diagnostic: labels are not used to fit clustering models, tune hyperparameters, or compute CHB descriptors. The goal is to quantify how strongly CVIs reflect class-induced structure rather than intrinsic, label-free clusterability.

CVI indices primarily assess global compactness and separation under implicit blob-like assumptions. Consequently, poor or moderate scores may arise from fundamentally different mechanisms: (i) genuine local ambiguity and neighborhood mixing (true separability collapse), (ii) topological mismatch due to non-convex or manifold-like structure, or (iii) scale and density heterogeneity across clusters. These mechanisms have distinct algorithmic implications, yet baseline metrics collapse them into a single scalar signal (and can even disagree across metrics, e.g., silhouette/DB vs. Hopkins in Table 32).

**Failure as a separability gate (A vs. not-A).**     MNIST and KMNIST provide decisive counterexamples. In raw pixel space, both exhibit slightly negative silhouette and elevated DB relative to clearly clusterable tabular datasets (Table 32), signals that often trigger a "collapse" diagnosis under common rules of thumb. However, CHB separation diagnostics reveal low neighborhood overlap, limited hub-mediated shortcutting, and robust inter-cluster margins, indicating that the separability gate *passes*. Their difficulty instead stems from pronounced Topology Mismatch (persistent non-convex structure), not from a lack of local evidence. Conversely, CIFAR-100 represents a genuine collapse case: extreme overlap and hubness invalidate local identifiability; baseline metrics correctly indicate severe difficulty (Table 32), but they do not isolate *why* the instance is hard.

**Failure as a post-gate discriminator (B vs. C).**     Baseline metrics also fail at the B vs. C boundary. Pendigits exhibits favorable silhouette and DB values relative to MNIST/KMNIST (Table 32), suggesting easier clustering. While this is directionally correct, baseline metrics cannot explain *why* Pendigits differs: CHB reveals that separability is strong and topology is simple, with remaining difficulty driven by multi-scale density and shape heterogeneity. Thus, baseline indices cannot distinguish topology-driven mismatch (Regime B) from scale-driven heterogeneity (Regime C), even when separability is usable in both cases.

**Implication (can one recover CHB regimes from the CVI?).**     Taken together, Table 32 shows that baseline clusterability metrics are insufficient for regime assignment: they fail both as a separability gate (A vs. not-A) and as a post-gate discriminator (B vs. C). Accordingly, one cannot recover CHB's mechanism-level regime structure from these baseline metrics alone (even when combined), because they do not directly measure local identifiability (overlap/hubness/margins) nor topology effects. This motivates CHB's separability-first design: we measure local identifiability directly and then attribute residual difficulty to cohesion heterogeneity or topology. Without this explicit decomposition, mechanism-aware evaluation and diagnosis are unreliable.

