# OpenReview forum: "CHB: A Diagnostic Toolkit for Hardness-Aware Clustering Evaluation"
_ICML.cc/2026/Conference — ICML 2026 regular_

### Official Review · Reviewer_V78c · 2026-03-11

**Soundness:** 3
**Presentation:** 2
**Significance:** 3
**Originality:** 3
**Overall Recommendation:** 4
**Confidence:** 3

**Summary:**

This paper proposes an interpretable hardness-aware clustering evaluation framework CHB to systematically explain algorithm behavior and representation effects. CHB contains three evaluation dimensions: separation, cohesion, and topology. The author uses CHB to conduct a comprehensive evaluation of some representative algorithm families and provides corresponding evaluation toolkits. The author puts forward some insightful conclusions through the evaluation.

**Compliance With Llm Reviewing Policy:**

Affirmed.

**Key Questions For Authors:**

1. Why does CHB use separation, cohesion, and topology to calculate a hardware fingerprint? Should feature relevance be considered? If it has been considered, which metric in CHB does it belong to?
2. The dataset used in this study is non-relational data. Clustering graph-structured data has also been widely studied in the field of clustering. Is CHB applicable to graph-structured data and graph clustering algorithms?
3. For the two improvement modes in section 6.2: separability rescue and mismatch correction, can you provide strict theoretical proof, or improve according to these modes based on a certain current SOTA algorithm, so as to verify whether it has practical guiding significance?

**Limitations:**

Yes

**Strengths And Weaknesses:**

Strengths

1. The article content is detailed, and the experimental design is sufficient.
2. The research question is very interesting and helpful to the field of clustering.
3. The proposed assessment framework may have practical implications

Weaknesses

1. The article is obscure and difficult to understand, making it difficult to understand some key claims.
2. Incomplete consideration of evaluation data

---

> ### Author Rebuttal · Authors · 2026-03-30
>
> We thank the reviewer for the constructive feedback. We address each concern below.
>
> ## Why These Three Axes? Feature Relevance
>
> We use separation, cohesion, and topology because they capture distinct clustering failure modes. Because CHB is a diagnostic, we use theory-grounded, interpretable axes over learned ones that may blur these mechanisms. Appendix G supports this choice: removing separation merges Regimes A/B, while removing topology merges Regimes B/C.
> We do not model feature relevance as a separate axis because it manifests through the existing fingerprint.  Irrelevant or noisy features weaken separability, distort cohesion, and increase apparent topological complexity. CHB therefore captures feature relevance indirectly through its effect on the dataset’s effective geometry.
>
>
> ##  Graph-Structured Data
>
> CHB assumes point-cloud data in a metric space. A graph extension is conceptually feasible—for example, separation via shortest-path distances, topology via edge-weight filtrations, and cohesion via graph-native analogs—but it requires dedicated validation on graph benchmarks.  We see opportunities for future work by combining CHB with Graph structured data.
>
>
> ##  Evaluation Breadth
>
> To broaden the stress test, the revision adds two text datasets, new tabular datasets, and a modern foundation-model pipeline.
>
> ### Regimes generalize across modalities
>
> To test whether the regime boundaries remain stable beyond the original benchmark, we applied CHB with frozen thresholds to new tabular and text datasets. Eucalyptus (tabular, n=641, d=19, K=5) fails the separability gate on all three indicators—the most severe gate failure in our expanded suite—and k-means achieves NMI = .08, consistent with Regime A. 20 Newsgroups (text, 20 classes) also lands in Regime A. AG News (text, 4 classes) lands in Regime B with usable separation but nontrivial topology. Cardiotocography (tabular, 10 classes) lands in Regime C with near-perfect separation and k-means NMI = .91. This extends CHB's stress testing to three modalities and shows that the regime logic generalizes across diverse data types.
>
> *The Table  1 below reports raw-feature regime assignments under frozen thresholds. The later CLIP→UMAP table reports regime trajectories after representation learning.*
>
> | **Regime** | **Image** | **Tabular** | **Text** |
> |---|---|---|---|
> | A | CIFAR-10/100, STL-10, SVHN | Eucalyptus | 20 Newsgroups |
> | B | MNIST, KMNIST, FMNIST | — | AG News |
> | C | — | Cardiotocography, Pendigits, ... | — |
>
> ---
>
> ## Stress-Testing with Modern Representations
>
> We also evaluated CHB on a modern pipeline (CLIP followed by UMAP). CHB reveals how each transformation affects different axes: the findings are described below.
>
> ### Image contrast: CIFAR-10 vs. SVHN
>
> Both datasets start in Regime A with nearly identical raw overlap. For CIFAR-10, CLIP rescues separation (A→B) and UMAP then collapses the remaining topology (B→C), raising k-means NMI from .08 to .80 (+.713). For SVHN, CLIP fails to rescue separation, so UMAP compresses already-overlapping classes and NMI drops from .080 to .036 (−.044). CHB therefore explains why the same pipeline helps CIFAR-10 but harms SVHN.
>
> ### Text data follows the same pattern
>
> AG News starts in Regime B; CLIP strengthens separation and UMAP yields B→C with +.192 NMI. 20 Newsgroups starts in Regime A; CLIP does not rescue separation across 20 fine-grained categories, and the pipeline remains trapped in the failure regime, with only a small net gain over raw features.
>
>
> *Table 2: CHB trajectories under the CLIP→UMAP pipeline are summarized below. "Sep. rescue?" refers to the effect of CLIP, "Trajectory" tracks regimes from raw features to the final embedding, and ΔNMI is the net change in k-means NMI over the full pipeline.*
>
> | **Dataset** | **Sep. rescue?** | **Trajectory** | **ΔNMI** |
> |---|---|---|---|
> | CIFAR-10 | yes | A→B→C | +.713 |
> | MNIST | already ok | B→C | +.347 |
> | AG News | strengthened | B→C | +.192 |
> | 20 NG | no | A→A | +.046 |
> | SVHN | no | A→A | −.044 |
>
> ### Mismatch correction at the clustering stage: Texture
>
> Texture has strong separation and simple topology, yet k-means scores only .64. CHB identifies cohesion heterogeneity as the bottleneck and points to full-covariance GMM. A one-line swap raises NMI from .64 to .90.
>
> These cases show why NMI alone is not enough: it reports the outcome, while CHB diagnoses the cause—failed separability, unresolved topology, or representation–objective mismatch. We position these modes as empirically validated regularities, not formal guarantees; we will clarify this boundary in the revision.
>
> ## Presentation Revisions
>
> We have improved the presentation for accessibility. A new workflow diagram traces the pipeline from raw data through the (S, C, T) fingerprint to regime assignment. The PCA projection into hardness space now appears in the main text with downstream NMI values, showing that the fingerprint separates easier from harder problems.

---

> > ### Author Rebuttal · Reviewer_V78c · 2026-04-02
> >
> > Although the author has provided simple ideas in response to Q2 regarding scaling on graph data, since this paper is a study evaluating clustering from a data perspective, it should cover a comprehensive range of data types to demonstrate the generalization capability of the proposed framework.

---

> > > ### Author Response · Authors · 2026-04-03
> > >
> > > We thank the reviewer for this helpful input and further suggestions.
> > > ## Generalization
> > > CHB targets point-cloud clustering in metric spaces, which covers many data types  of clustering benchmarks in the literature. Our frozen-threshold regime assignments generalize across tabular, image, and text data without retuning, and the CLIP→UMAP experiments further demonstrate CHB's diagnostic value for modern representation pipelines — together providing strong evidence of structural robustness within this scope.
> > > ## Graph data
> > > We agree that extending CHB to graph-structured data is an important next step. The core idea of CHB—decomposing clustering hardness into interpretable axes and mapping datasets to structural regimes—appears conceptually transferable beyond the current setting. In the present paper, we provide evidence for this diagnostic logic within the scope we study, namely point-cloud data in metric spaces, where CHB remains informative across different data types.  We will therefore expand the limitations section to state this scope boundary more clearly and to outline graph-native extensions as a concrete future direction.

---

### Official Review · Reviewer_A4fE · 2026-03-13

**Soundness:** 3
**Presentation:** 3
**Significance:** 3
**Originality:** 3
**Overall Recommendation:** 5
**Confidence:** 3

**Summary:**

Different benchmark datasets can present different challenges for clustering, depending on their geometry. It is important to take these challenges into account when comparing clustering algorithms’ performance.


This paper provides a suite of dataset-specific statistics which can reveal these challenges. The three main challenges that these statistics quantity are: separation (do local neighborhoods suggest clusters?), cohesion (do clusters share similar properties?), and topology (does high-order connectivity like loops complicate recovery?).


They show that CHB enables a more fine-grained understanding of cluster benchmarks: revealing distinct “regimes” in this space of statistics, where different clustering algorithms and learned representations display qualitatively different strengths and weaknesses on a suite of real datasets.

**Compliance With Llm Reviewing Policy:**

Affirmed.

**Key Questions For Authors:**

I can see why the topology signature probes for connected components and loops. However I find it curious that it would probe for voids (PH2) and stop there. Surely real high-dimensional datasets feature higher-order topological structure? These structures become less “intuitive” to us, but maybe they still could affect a clustering algorithm’s performance? It doesn’t seem unreasonable that the potential impact could scale with the complexity of the topology.


To what extent have you stress-tested this model? What would you consider the weaknesses? Are there datasets you have encountered or can cook up which don’t quite get the right treatment from these diagnostics. I can imagine, based on the previous question, that datasets with higher-order topological


I think there is much to be understood regarding the possibilities of a learned representation “rescuing separability” in a dataset. Do you have a sense of the limits here? It may be useful to run more experiments, say on synthetic data, to see where these embedding techniques break down (and which technique is the most robust).

**Limitations:**

yes

**Strengths And Weaknesses:**

(Originality) I like this paper insofar as it presents a very simple and appealing idea –– a taxonomy of datasets according to their clustering-relevant geometry –– that has not seemed to have made its way into the standard practice of clustering. I am not aware of significant previous work that has attempted to do this. To my knowledge, most theory work on clustering restricts attention to specific clustering objectives like k-means, which essentially defines clusters as well-separated spherical blobs. The notable exception to this is spectral clustering, which leverages clustered neighborhood structure to convert non-spherically-clustered data into a spherical-blobs representation.

Speaking of which, the paper should make this distinction about spectral clustering clear. It should not be considered a separate suite of methods as classical-center algorithms; it should be thought of as a classical-center algorithm applied to a well-understood learned representation.In light of this, I think the “Representation Auditing” section should be updated accordingly, with the spectral clustering embedding (“Laplacian eigenmaps” as it is sometimes referred to) as one of the main learned representations of interest. The authors might also consider embedding methods like VAEs with various architectures, t-SNE, Kamada-Kawai MDS, hubness reduction procedures like ICDM, etc.

(Presentation) The distinction between Regimes A, B, and C are intuitive and interesting, though I think it’s easy to get lost in the details in the current way it is presented––especially the discussion of these various thresholds in page 5, left column. I think this paper could benefit from a “headline figure,” say at the top of the second page, which illustrates these conceptual differences.

(Significance) I could imagine interesting follow-ups to this work. On the practical side, I think this framework ought to be incorporated into more cluster benchmark studies and kept in mind when analyzing learned embeddings. On the theory side, I think this idea of a cluster diagnostic could be formalized and refined to answer questions like: to what extent can a learned representation of a given dataset move my position in cluster diagnostic space? I think this is understood relatively well for spectral clustering, but not for many other learned representations.

(Soundness) My main worry here is that the model has not been stress-tested enough. I’m not sure we can rule out the possibility of a (realistic, not just pathological) dataset which scores well on the CHB diagnostics but eludes all existing clustering methods.

---

> ### Author Rebuttal · Authors · 2026-03-30
>
> We thank the reviewer for the positive assessment and constructive suggestions.
>
> ##  Stress-Testing — Regime Breadth and Failure Cases
>
> To broaden the stress test, the revision adds two text datasets, new tabular datasets, and a modern foundation-model pipeline (detailed below), all evaluated with frozen thresholds and no re-tuning.
>
> ### Regimes generalize across modalities
>
> To test whether the regime boundaries remain stable beyond the original benchmark, we applied CHB with frozen thresholds to new tabular (Eucalyptus, Cardiotocography) and text datasets (20 Newsgroups, AG News). Eucalyptus (tabular, n=641, d=19, K=5) fails the separability gate on all three indicators—the most severe gate failure in our expanded suite—and k-means achieves NMI = .08, consistent with Regime A. 20 Newsgroups (text) also lands in Regime A. AG News (text) lands in Regime B with usable separation but nontrivial topology. Cardiotocography (tabular) lands in Regime C with near-perfect separation and k-means NMI = .91. This extends CHB’s stress testing to three modalities and shows that the regime logic generalizes across diverse data types.
>
> The table below reports raw-feature regime assignments under frozen thresholds. The later CLIP→UMAP table reports regime trajectories after representation learning.
>
> | Regime | Image | Tabular | Text |
> |---|---|---|---|
> | A | CIFAR-10/100, STL-10, SVHN | Eucalyptus | 20 Newsgroups |
> | B | MNIST, KMNIST, FMNIST | — | AG News |
> | C | — | Cardiotocography, Pendigits, ... | — |
>
> ## Stress-Testing with Modern Representations
>
> We evaluate CHB on a CLIP→UMAP pipeline, a two-stage pipeline whose components affect different dimensions of clustering hardness in our experiments, and ask whether regime assignments can identify which stage is likely to help clustering.
>
> The two stages target different CHB axes. In our experiments, CLIP primarily improves separability, while UMAP mainly performs mismatch correction by simplifying representation geometry.  UMAP cannot recover separability when CLIP fails to provide it. CHB therefore explains not just whether the pipeline helps, but why, by distinguishing gains from separability rescue from gains due to geometric simplification.
>
> ### Image contrast: CIFAR-10 vs. SVHN
>
> Both datasets start in Regime A with nearly identical raw overlap. For CIFAR-10, CLIP rescues separation (A→B) and UMAP then collapses the remaining topology (B→C), raising k-means NMI from .08 to .80(+.713). For SVHN, CLIP fails to rescue separation, so UMAP compresses already-overlapping classes and NMI drops from .080 to .036(-.044). CHB therefore explains why the same pipeline helps CIFAR-10 but harms SVHN.
>
> ### Text data follows the same pattern
>
> AG News starts in Regime B; CLIP strengthens separation and UMAP yields B→C with +.192 NMI. 20 Newsgroups starts in Regime A; CLIP does not rescue separation across 20 fine-grained categories, and the pipeline remains trapped in the failure regime, with only a small net gain over raw features. Together, these cases show that CHB remains informative on modern learned representations across modalities.
>
>
> **Table 1:** CHB trajectories under the CLIP→UMAP pipeline. *Sep. rescue?* indicates whether CLIP improves separation, *Trajectory* tracks regimes from raw features to the final embedding, and ΔNMI denotes the net change in *k*-means NMI across the full pipeline.
>
> | Dataset  | Sep. rescue? | Trajectory | ΔNMI  |
> |----------|--------------|------------|------:|
> | CIFAR-10 | yes          | A→B→C      | +.713 |
> | MNIST    | already ok   | B→C        | +.347 |
> | AG News  | strengthened | B→C        | +.192 |
> | 20 NG    | no           | A→A        | +.046 |
> | SVHN     | no           | A→A        | -.044 |
>
> These cases show why NMI alone is not enough: it reports the outcome, while CHB diagnoses whether the bottleneck is failed separability or unresolved topology.
>
> ## Spectral clustering as a learned representation
>
> We agree that spectral clustering is best viewed as a classical-center method on a learned representation. CHB naturally captures this: on MNIST, the spectral embedding shifts the fingerprint from B→C—the same trajectory observed for UMAP and DDC—showing that CHB's diagnostics generalize across embedding families.
>
> ## Higher-order topology beyond PH2
>
> In the current experiments, PH0–PH2 provides enough topological resolution for CHB. It cleanly separates datasets with substantial topological structure from those without and supports the regime distinctions observed across the benchmark.  Higher-order topology fits naturally into the CHB framework, but it comes with substantially higher computational cost and did not appear necessary in our current benchmark.  We therefore see PH3+ as a natural extension rather than a necessary part of the current benchmark.
>
>
> ### Visualization
> A new workflow diagram and the PCA hardness-space projection (now in the main text, with NMI values) together improve the immediate readability of the regime structure.

---

> > ### Author Rebuttal · Reviewer_A4fE · 2026-04-03
> >
> > Thank you for the detailed rebuttal. Some further comments:
> >
> > - I like the follow-up experiments. In addition to this "stress testing" I think including some "sanity checks" would be very useful for this paper. For instance, the half-moons or concentric circles dataset is canonically "rescued" by spectral clustering, while high-dimensional Gaussian data with a few "hubs" in the middle could be "rescued" by a hubness reduction procedure like ICDM (see https://www.jmlr.org/papers/volume11/radovanovic10a/radovanovic10a.pdf). Showcasing these examples in the paper itself could really improve the presentation and the message.
> >
> > - I've stumbled upon this reference which strikes me as quite relevant. It suggests developing a "taxonomy of clustering problems" and discusses some difficulties inherent to such an endeavor. https://proceedings.mlr.press/v27/luxburg12a/luxburg12a.pdf
> >
> > I maintain my score.

---

> > > ### Author Response · Authors · 2026-04-03
> > >
> > > We thank the reviewer for the positive assessment and the thoughtful follow-up suggestions.
> > >
> > >
> > > ## Sanity-check subsection
> > > We especially appreciate the concrete proposals on sanity checks and related work.
> > > We agree that canonical sanity-check examples would strengthen both the exposition and the paper's message. We plan to add a small sanity-check subsection so that readers can connect the hardness fingerprint to these familiar clustering pathologies directly.
> > >
> > > ## Related work Luxenburg (2012)
> > > We also appreciate the pointer to von Luxburg et al. (2012). Their central argument — that clustering evaluation cannot be divorced from end-use context — is an important framing for our work. CHB does not claim to resolve this context-dependence; rather, it operates within the most common evaluation setting in the current literature: external evaluation against ground-truth labels. Even within this fixed protocol, datasets vary in structural difficulty in ways that leaderboard scores obscure, and CHB adds interpretability to that setting. We will discuss this connection in the related work section, positioning CHB as addressing a complementary gap — making the dominant evaluation paradigm more transparent — rather than as a general-purpose answer to the broader taxonomy program they outline.

---

### Official Review · Reviewer_uwaD · 2026-03-13

**Soundness:** 2
**Presentation:** 2
**Significance:** 2
**Originality:** 3
**Overall Recommendation:** 3
**Confidence:** 4

**Summary:**

This paper suggests a collection of measures quantifying various aspects of a data set capturing how difficult it is to successfully cluster. This diagnostic toolkit is intended to help evaluation of clustering methods, by providing insight into why such methods may fail or succeed on a given data set.

The diagnostic toolkit is called “CHB”, but strangely this acronym seems not to be defined in the title, abstract or introduction of the paper, so I don’t know what it is intended to convey.

**Compliance With Llm Reviewing Policy:**

Affirmed.

**Final Justification:**

As per my rebuttal acknowledgement, in my view the revisions have improved the presentation, but only to a limited extent.

**Key Questions For Authors:**

I have no questions to ask.

**Limitations:**

There is some discussion of limitations in appendix K, at the very end of the supplementary material. I think the paper would be better balanced by these appearing in the main part of the paper, where they are much more likely to be read.

**Strengths And Weaknesses:**

In terms of soundness, the overall aim of providing some kind of quantitative diagnostic to help investigate clustering seems sensible. The constituent parts of the “fingerprint” are described with little mathematical detail so I can’t be certain of their soundness. I’m very familiar with k-nn graphs, persistent homology etc., but the descriptions in section 3.2 are largely verbal leaving many mathematical details unclear (as two illustrative examples: what is the value of k in the in the k-nn graph? What is the scale considered persistent homology?).

Considering the emphasis on a multi-dimensional representation of a data set in terms of the metrics, I was hoping to see some revealing visualisations of where various data sets fall in this multidimensional space. This feels like strange omission to me, and without that kind of visualisation I think it’s not easy to comment on the soundness of the metrics. The additional numerical results in the appendices did not help me.

The presentation of the paper seems very long-winded to me, I think the essential aspects of the metrics and an effective summary of background could be achieved in 1-2 pages, but it is drawn out to 4 pages in the paper. As such I think the reader is likely to lose interest before getting to the core content. I didn’t find the question-and-answer style of the sections 5 and 6 informative, and it leads to overcomplicated section numbering (e.g., “5.1. Q2.2”, etc.)

The constituent metrics are not mathematically new, so I suppose it is the combination of them which is original.

I imagine that researchers would be potentially keen to embrace some new clustering diagnostic if it provides novel and useful insights. The overall weakness of this paper is that the presentation is so long-winded and vague that I think it is unlikely that researchers would engage with it.

---

> ### Author Rebuttal · Authors · 2026-03-30
>
> We thank the reviewer for the careful reading. The revision addresses the three dimensions raised in the review: (i) mathematical specification, (ii) visualization of the hardness space, and (iii)  readability. It also broadens the empirical evaluation with new datasets and modern representation pipelines.
>
> **Key changes at a glance.**
>
> - CHB is defined at first use as *Clustering Hardness Benchmark* in the abstract, and introduction;
> - a formula-and-parameter table specifies all quantities in the fingerprint;
> - two overview figures are added to the main text to visualize the hardness space;
> - the Q1/Q2/Q3 organization is replaced with standard sectioning and flat numbering;
> - the limitations discussion is moved from the appendix into the main paper;
> - the empirical scope is expanded to text, tabular, and learned-representation pipelines.
>
> ---
>
> ## 1. Mathematical Specification
>
> The revision makes CHB explicit as a hardness fingerprint
>
> $$h(\mathcal{D}) = (S, C, T),$$
>
> where *S* summarizes separability, *C* summarizes cohesion heterogeneity, and *T* summarizes topology. Each component is now introduced with: (i) a one-line intuition, (ii) its formal definition, and (iii) its parameterization. The specific gaps noted in the review are now stated directly:
>
> - the *k*-NN graph is evaluated at *k* ∈ {10, 20, 40}, with stability checked across this range;
> - persistent homology is computed with landmark subsampling at *m*₀ = 800, *m*₁ = 400, *m*₂ = 384, and filtration tolerance *τ* = 0.01;
> - the separability gate threshold *τ*_TOP = 15.0 is calibrated from the 95th percentile of a Gaussian-blob null model.
>
>
> ---
>
> ## 2. Visualizing the Hardness Space
>
> The revision adds two figures to the main text. A new workflow figure on page 1 shows the full pipeline from raw data to the (*S*, *C*, *T*) fingerprint, regime assignment, and diagnostic interpretation. The PCA projection of all datasets in CHB space, previously in the appendix, is now promoted to the main text and colored by downstream NMI. Together, these make the regime structure visible much earlier and let the reader directly inspect whether the fingerprint separates easier from harder clustering problems.
>
> ---
>
> ## 3. Structure and Readability
>
> The paper no longer uses the Q1/Q2/Q3 format, and nested numbering such as "5.1. Q2.2" has been removed. The revised main sections are flatter and standard. The limitations discussion is also moved into the main text.
>
> ---
>
> ## 4. Two Examples of What CHB Adds Beyond NMI
>
> CHB combines separability, cohesion heterogeneity, and topology into a single fingerprint that distinguishes distinct clustering failures—something no single score can do. The ablation in Appendix G confirms all three axes are necessary: dropping separation merges Regimes A/B; dropping topology merges B/C.
>
> The revision now includes case studies showing this compositional value in two settings.
>
> **Algorithm selection on raw features.** When clustering is applied directly to raw features, CHB identifies which structural bottleneck to address.
>
> | Dataset | CHB diagnosis | Guided fix | NMI |
> |---------|--------------|------------|-----|
> | Texture | cohesion mismatch | *k*-means → full-cov. GMM | .64 → .90 |
>
> Texture has strong separation and simple topology, yet *k*-means scores only .64. CHB identifies cohesion heterogeneity as the bottleneck—not lack of cluster signal—and points to a flexible covariance model. A one-line swap to full-covariance GMM raises NMI to .90.
>
> **Representation pipeline auditing.** When modern representations are applied before clustering, CHB tracks *which stage* succeeds or fails—something a final NMI value alone cannot reveal. The revision adds two text datasets, new tabular datasets, and a modern CLIP→UMAP pipeline, all evaluated with frozen thresholds.
>
> | Dataset | Sep. rescued by CLIP? | Trajectory | NMI |
> |---------|-----------------------|------------|-----|
> | CIFAR-10 | ✓ | A→B→C | .08 → .80 |
> | SVHN | ✗ | A→A | .08 → .04 |
>
> CIFAR-10 and SVHN start in the same regime with near-identical raw overlap. Under the same CLIP→UMAP pipeline, CIFAR-10 improves dramatically while SVHN gets worse. CHB explains the divergence: CLIP rescues separability for CIFAR-10, enabling UMAP to simplify the remaining geometry; for SVHN, CLIP fails to rescue separability, so UMAP compresses already-overlapping classes. Both distinctions—algorithm mismatch vs. representation failure, successful vs. failed pipeline—are invisible to a leaderboard score but immediate in the fingerprint.

---

> > ### Author Rebuttal · Reviewer_uwaD · 2026-04-02
> >
> > Thanks to the authors for their detailed response and revisions which I believe improve presentation somewhat, so I have increased my score a little.

---

> > > ### Author Response · Authors · 2026-04-03
> > >
> > > We thank the reviewer for the careful reading and the concrete suggestions on improving the paper — these directly shaped the overall paper presentation. We are glad the changes addressed the concerns raised.

---

### Official Review · Reviewer_RTqA · 2026-03-15

**Soundness:** 3
**Presentation:** 3
**Significance:** 3
**Originality:** 3
**Overall Recommendation:** 4
**Confidence:** 3

**Summary:**

The papers makes contributions in the area of evaluation methods for clustering. Contrary to many prior approaches, that summarize the performance of clustering methods across a variety of benchmarks and datasets using a single score, this paper proposes a thorough examination of the different aspects that change the behavior of clustering algorithms. For example, different datasets may be characterized by different assumptions on their geometry and their connectivity, the types of underlying ground-truth clusters, their cohesion properties and well-separatedness properties, and as a result, just having a single ranking over different clustering methods can often be misleading.

The paper proposes CHB which is a diagnostic framework for evaluation purposes. At its heart, CHB computes a hardness fingerprint embedding, so that each dataset is mapped into point in an interpretable space spanned by three axes: separability, cohesion heterogeneity, and topological structure. These three axes are typically properties that have been studied in the literature and are often found to be the reason why certain clustering tasks are easier or harder. CHB helps interpreting external evaluation outcomes based on these hardness coordinates and thus helps transform the leaderboard scores into more meaningful explanations.

For the experiments, the authors used traditional image benchmarks (MNIST, KMNIST, FMNIST, CIFAR-10/100, STL10, SVHN) and tabular datasets from Asuncion et al. 2007.

**Compliance With Llm Reviewing Policy:**

Affirmed.

**Final Justification:**

I like this paper and believe it to be a nice framework for clustering evaluation. I maintain my positive score after the rebuttal.

**Key Questions For Authors:**

See Weaknesses above.

**Strengths And Weaknesses:**

Strengths:

- CHB seems to be a useful tool to explain why certain clustering tasks have been easier or harder

- CHB can be useful in the industry as an analysis tool for a specific dataset

- The main finding (refering to Q1) that most clustering benchmarks can indeed be organized into a small number of interpretable hardness regimes is quite interesting.

Weaknesses:

- Though the three axes are informed from clustering theory, perhaps there are other variables that explain the clustering benchmarks. Have you tried learning those parameters instead of pre-deciding what is the important property that makes the task harder or easier?

- The datasets used thought traditional and standard are quite small for today's standards. I was expecting a more in-depth evaluation across larger datasets for an evaluation paper on clustering.

- Even though helpful for diagnostic purposes post-hoc, I believe most applications would benefit from a prognostic tool that predicts which clustering methods should perform best for the dataset at hand. Can you use findings from CHB to analyze a dataset across the 3 axes or others, perhaps on a subsampled dataset, and help predict which method would be best?

---

> ### Author Rebuttal · Authors · 2026-03-30
>
> We thank the reviewer for the constructive feedback on three concerns: (i) whether theory-grounded axes are preferable to learned ones, (ii) whether CHB remains informative on broader datasets and modern pipelines, and (iii) whether CHB can guide method selection.
>
> ---
>
> ## Learned vs. Pre-Specified Axes
>
> The reviewer asks whether other variables might better explain clustering difficulty. We chose theory-grounded axes because CHB is a *diagnostic*: each axis maps to a distinct failure mode (Table 1 from paper) and a distinct intervention. Learned axes may recover predictive power but risk collapsing distinct failure modes into an opaque score, weakening the ability to say *why* a method fails and *what to change*.
>
> Appendix G supports this: removing separation merges Regimes A/B; removing topology merges B/C. The (*S*,*C*,*T*) fingerprint is the minimal interpretable set preserving regime boundaries.
>
> We do not view learned selectors as competing with CHB. A natural extension is a learned predictor *on top of* the fingerprint—interpretable axes as state description, learned selector for finer-grained ranking. We will clarify this in the revision.
>
> ---
>
> ##  Broader Evaluation and Modern Pipelines
>
> We have extended the benchmark by  text (AG News, 20 Newsgroups) datasets, tabular (Eucalyptus, Cardiotocography) datasets, and a CLIP→UMAP pipeline, all with **frozen thresholds** and no re-tuning.
>
> ### Regimes generalize across modalities
>
> Eucalyptus (tabular, *n*=641, *d*=19) fails the separability gate on all three indicators and *k*-means achieves NMI = .08, consistent with Regime A. 20 Newsgroups (text, 20 classes) also lands in A. AG News (text, 4 classes) lands in B. Cardiotocography (tabular, 10 classes) lands in C with NMI = .91. The regime structure transfers across three modalities without modification.
>
> | **Regime** | **Image** | **Tabular** | **Text** |
> |---|---|---|---|
> | A | CIFAR-10/100, STL-10, SVHN | Eucalyptus | 20 Newsgroups |
> | B | MNIST, KMNIST, FMNIST | — | AG News |
> | C | — | Cardiotocography, Pendigits, … | — |
>
> ### Stress-testing with modern representations
>
> We evaluate CHB on a CLIP→UMAP pipeline and ask whether regime assignments can identify which stage is likely to help before full downstream clustering is run. In our experiments, the two transformations induce distinct shifts in the CHB profile: CLIP primarily rescues separability, whereas UMAP mainly performs mismatch correction by simplifying representation geometry.
> Tracking the CHB profile after each stage makes these roles explicit. UMAP does not create separability when CLIP fails to provide it. CHB therefore helps explain not just whether the pipeline succeeds, but why, by separating improvements due to increased separability from those due to geometric simplification. Below we outline this in detail.
>
> **Image: CIFAR-10 vs. SVHN.** Both start in Regime A. CLIP rescues CIFAR-10's separation (A→B), UMAP collapses topology (B→C), *k*-means rises from .08 to .80. For SVHN, CLIP fails to rescue separation; UMAP compresses already-overlapping classes and NMI drops from .080 to .036. Without CHB, a practitioner must run full clustering to discover this; with CHB, the failed gate after CLIP already signals it.
>
> **Text: AG News vs. 20 Newsgroups**. The text datasets show the same stage-specific behavior. AG News starts in Regime B and moves to C, with a +.192 NMI gain. 20 Newsgroups starts in Regime A and remains trapped there, indicating that CLIP is unable to rescue separability across its 20 fine-grained categories.
>
> | **Dataset** | **Sep. rescue?** | **Trajectory** | **ΔNMI** |
> |---|---|---|---:|
> | CIFAR-10 | yes | A→B→C | +.713 |
> | MNIST | already ok | B→C | +.347 |
> | AG News | strengthened | B→C | +.192 |
> | 20 NG | no | A→A | +.046 |
> | SVHN | no | A→A | −.044 |
>
> Across all cases, the CHB regime after CLIP correctly signals whether the rest of the pipeline is likely to help. While NMI reports only the endpoint, CHB provides an earlier and more informative diagnosis by identifying the structural bottleneck before the final clustering stage.
>
> ---
>
> ## Toward Prognostic Use
>
> CHB can guide method selection, not just explain outcomes post-hoc. CHB is diagnostic by design, but the regime assignments offer initial prognostic value by narrowing the plausible intervention *class* before clustering is run:
>
> 1. **Separability gate fails → consider representation learning first.** SVHN and 20 Newsgroups illustrate: no downstream objective recovers from absent separation.
> 2. **Separation usable, topology complex → topology-simplifying embeddings are a natural next step.** The CIFAR-10 and AG News B→C transitions follow this pattern.
> 3. **Topology simple, cohesion heterogeneity high → flexible covariance models may help.** Texture illustrates: full-covariance GMM raises NMI from .64 to .90.
>
>
> **Beyond diagnosis.** SVHN,  and 20 Newsgroups remain in Regime A even under modern representations, exposing failures that averaged leaderboards hide.

---

> > ### Author Rebuttal · Reviewer_RTqA · 2026-04-01
> >
> > Thank you for your responses! I maintain my original score for the present version of the paper!

---

> > > ### Author Response · Authors · 2026-04-03
> > >
> > > We thank the reviewer for the constructive evaluation and appreciate the confirmation that the rebuttal addressed the main concerns. The cross-modality experiments and the CLIP→UMAP pipeline analysis will appear in the revised version, strengthening the evaluation breadth of the paper.

---

### Decision · Program_Chairs · 2026-04-30

**Decision:**

Accept (regular)

**Comment:**

This paper presents a diagnostic framework for clustering evaluation that attempts to encompass a more nuanced, interpretive understanding of clustering difficulty called CHB. Reviewers lean positive though they mention that the exposition is verbose and drawn out at parts. They mention the conceptual strength and originality of mapping datasets into an interpretable “hardness fingerprint” that is intuitively defined by separability, cohesion heterogeneity, and topology. The empirical finding that widely used clustering benchmarks naturally organize into a small number of interpretable hardness regimes is interesting in its own right. There are some issues in the experiments that some reviewers note they feel will be resolved based on the progress during the rebuttal. Taken together, the reframing of clustering evaluation is offered through a principled lens and the paper makes a meaningful contribution to the area of clustering evaluation. Please continue to incorporate the constructive suggestions from the reviewers in preparing the revised version